# Magnetic and mechanical hardening of nano-lamellar magnets using thermo-magnetic fields

Liuliu Han ◉[1] ✉, Jin Wang[2], Nicolas J. Peter ◉[2], Fernando Maccari ◉[3], András Kovács ◉[4], Ruth Schwaiger ◉[2], Oliver Gutfleisch[3] & Dierk Raabe ◉[1]

High-performance magnetic materials based on rare-earth intermetallic compounds are critical for energy conversion technologies. However, the high cost and supply risks of rare-earth elements necessitate the development of affordable alternatives. Another challenge lies in the inherent brittleness of current magnets, which limits their applications for high dynamic mechanical loading conditions during service and complex shape design during manufacturing towards high efficiency and sustainability. Here, we propose a strategy to simultaneously enhance the magnetic and mechanical performance of a rare-earth-free multicomponent magnet. We achieve this by introducing nano-lamellar structures with high shape anisotropy into a cobalt–iron–nickel–aluminum material system through eutectoid decomposition under externally applied thermo-magnetic fields. Compared to the conventional thermally activated processing, the thermo-magnetic field accelerates phase decomposition kinetics, producing finer lamellae spacings and smaller eutectoid colonies. The well-tailored size, density, interface, and chemistry of the nano-lamellae enhance their pinning effect against the motion of both magnetic domain walls and dislocations, resulting in concurrent gains in coercivity and mechanical strength. Our work demonstrates a rational pathway to designing multifunctional rare-earth-free magnets for energy conversion devices such as high-speed motors and generators operating under harsh service conditions.

Magnetic materials are key components for the green energy transition due to their role in sustainable wind energy conversion, electromobility, automation, and robotics[1] and provide stable high torque in motor applications[2]. They can be classified based on their coercivity and energy products into soft, semihard, and hard magnetic materials. The critical rare earth (RE) elements[3–6] are generally required in magnetically hard materials to provide strong spin-orbit coupling for achieving high magnetocrystalline anisotropy, which is the basis for large magnetic hysteresis[7]. Alternative methods for enhancing anisotropy include introducing microstructural features with high shape anisotropy[8], stress anisotropy[9–11], chemical anisotropy[12,13], and other ways to enhance magnetocrystalline anisotropy[4]. In addition, extensive efforts in designing high-performance magnetic materials have been made to mitigate the ever-increasing requirement for applications under harsh environments, including high temperature, corrosion, hydrogen embrittlement, and mechanical loading conditions for sustainable electrification[14–17]. For example, hard magnets with good mechanical properties are required in the advanced

[1]Max Planck Institute for Sustainable Materials, Max-Planck-Straße 1, Düsseldorf, Germany. [2]Institute of Energy Materials and Devices, Forschungszentrum Jülich, Jülich, Germany. [3]Institute of Materials Science, Technical University of Darmstadt, Darmstadt, Germany. [4]Ernst Ruska-Centre for Microscopy and Spectroscopy with Electrons, Forschungszentrum Jülich, Jülich, Germany. ✉e-mail: l.han@mpie.de

flywheel energy storage systems that target an ultra-fast rotating speed of over 50,000 revolutions per minute for high-efficiency and sustainable electrification. This is because the rotating generates high dynamic mechanical and cyclic stress, and almost all conventional hard magnets fail at such speeds, thus limiting the efficiency of the whole system. However, most commercial hard magnetic materials are based on intermetallic compounds and are thus inherently brittle, a feature that renders them incapable of bearing plastic deformation during processing and service[18–20]. Therefore, the strongest hard magnets, e.g., Nd–Fe–B[21–24] and Sm–Co[25–27] alloys, are generally synthesized by powder metallurgy, which consists of a series of energy-, time-consuming and health-risky processing steps, i.e., metal powder production, mixing, compacting and sintering, as compared to the conventional casting and forging. Although near-net shape manufacturing can reduce the need for machining and the loss of materials, high-performance magnetic materials manufactured by adjustable processing with the potential to fill the cost and performance gap of the current magnetic systems are required for efficient and sustainable electrification. The current methods for manufacturing magnets still face significant challenges, particularly in minimizing material waste and ensuring product quality[28]. This is because the properties of magnets are highly relative to the microstructure, which in turn is dependent on the manufacturing process. The inherent brittleness of intermetallic compounds is prone to form microcracks during rapid cooling, which increases the risk of fracture during subsequent machining. For instance, Sm-Co-based alloys have a high machining failure of ~20% because of their low fracture toughness (1.9 ~ 2.9 MPa $m^{1/2}$)[28,29]. More specifically, this material design challenge is characterized by the need for a multifunctional profile, including high remanence and coercivity; good mechanical performance, e.g., high hardness, yield strength, ductility, and damage tolerance as well as good oxidation and corrosion resistance for applications under harsh chemo-mechanical operating conditions.

To overcome these design challenges, multicomponent materials (MCAs), which provide a wide range of chemical compositions and microstructures, thus leveraging a certain potential for realizing multifunctionality, have been proposed[30–32]. MCAs can reconcile different magnetic and mechanical features in one material because continuous chemical adjustment allows chemical adjustment within a continuous phase space into regimes where magnetic properties (such as Curie temperature, coercivity, and saturation magnetization) and mechanical properties (such as hardness and yield strength) can be simultaneously enhanced[33–35]. In addition, the option for possibly also chemically decorating and functionalizing microstructural defects at different length scales, e.g., dislocations, interfaces, and precipitates, can be exploited to help pin the movement of magnetic domain walls during magnetization and demagnetization, thus leading to a larger hysteresis integral[12,36,37]. The enhanced internal stress level due to the introduction of microstructural defects also increases the mechanical strength by impeding dislocation movement.

We translated these mutually conflicting design criteria into a material design strategy by triggering the formation of ferromagnetic nano-lamellae with high shape anisotropy via thermo-magnetic annealing. We selected a $Co_{28.6}Ni_{28.8}Fe_{30.6}Al_{12.0}$ (at.%) MCA system as a basis for this material development project. The ferromagnetic 3 d transition elements, i.e., Co, Ni, and Fe, are blended in a near-equiatomic ratio. They form a solid solution matrix with high crystal symmetry (face-centered cubic: fcc) due to its broad solubility range, thus enabling a ductile plastic response and leveraging high spontaneous magnetization. The addition of Al with a large atomic size and negative enthalpy of mixing with Co, Ni, and Fe triggers the formation of precipitates, thus enhancing coercivity and mechanical strength. This enables the hierarchical precipitation reactions, which are all coupled with tunable magnetic properties[38–42].

More importantly, we additionally led the alloy through a processing route that allowed us to harness the nano-lamellar microstructure design during eutectoid decomposition under an external thermo-magnetic field. The thermo-magnetic annealing (TMA) conditions were chosen based on several considerations: (a) the annealing temperature was placed in the eutectoid decomposition region; (b) the temperature was lower than the Curie temperature of the decomposed ferromagnetic phases during TMA; and (c) the applied maximum possible magnetic field, i.e., 9 Tesla (T), guarantees the alignment of all magnetic moments in the ferromagnetic phases along the magnetic field direction. This drives atomic diffusion and potentially changes the crystallinity and anisotropy, thus leading to improved magnetic performance[43–46]. To date, limited research has been conducted on the underlying mechanisms of the effect of TMA processing on the microstructure and associated magnetic and mechanical properties of MCAs[47]. In addition, the anisotropic counterparts containing aligned grains and texture induced by mechanical rolling can tune the kinetics of the subsequent phase decomposition of the current alloy system. This indicates the possibilities of further tailoring the shape anisotropy and chemistry of the nano-lamellae structure towards an enhanced magnetic and mechanical performance.

Here, we show a material design strategy that unlocks uncharted multifunctional processing and property terrain for engineering mechanically and magnetically strong magnets. This allows such magnets to withstand severe mechanical loading conditions during service as load-bearing components and provides an additional processing window to enhance induced magnetic anisotropy by plastically deforming the material, whereas conventional hard magnetic materials are limited due to their brittleness. This strategy can also be applied to other magnetic materials that tolerate inelastic loading without catastrophic failure during manufacturing, such as the Mn-Al system[48].

## Results and discussion
### Alloy design and microstructure analysis
First, we performed thermodynamic simulations (Methods) to identify a phase with high magnetization and high shape anisotropy in the $Co_{28.6}Ni_{28.8}Fe_{30.6}Al_{12.0}$ (at.%) MCA system. The simulation results (Supplementary Fig. 1) show that a two-phase regime comprising fcc and body-centered cubic (bcc) phases forms in the temperature range from 923 to 1623 K. The fcc phase further decomposes via a eutectoid phase transformation below 923 K and therefore has the potential to form a lamellar structure with high aspect ratio and shape anisotropy. Building on this simulation, we synthesized the material by conventional vacuum induction melting and conducted a two-stage isothermal heat treatment, i.e., first, homogenization at 1473 K for 2 h to obtain a fcc phase in a high volume fraction ( ~ 85 vol.%), and then, thermal annealing (TA) at 873 K targeting a eutectoid lamellar microstructure that can leverage both mechanical and magnetic hardening. This is because the lamellae and associated high interfacial density enhance their pinning effect against the motion of both magnetic domain walls and dislocations. Considering the potential effect of the external thermo-magnetic field on the kinetics of phase decomposition, the material variants were processed by TMA under a 9 T magnetic field at 873 K for 1–10 h, as shown in the schematic illustration in Fig. 1a. More specifically, the external thermo-magnetic field aligns all magnetic moments in the ferromagnetic phases during annealing. This enables tailoring of the crystallinity, crystallographic texture, and chemistry of the ferromagnetic phases toward high magnetization, high interface density, small lamellae spacing, and small colony size for mechanical and magnetic hardening[43–46,49,50].

The microstructure, magnetic, and mechanical properties of the MCAs obtained by the TMA and TA procedures were then investigated in detail. Ex-situ electron backscatter diffraction (EBSD) and electron channeling contrast imaging (ECCI) analyses reveal eutectoid decomposition during TMA (Fig. 1b, c). The decomposition area gradually

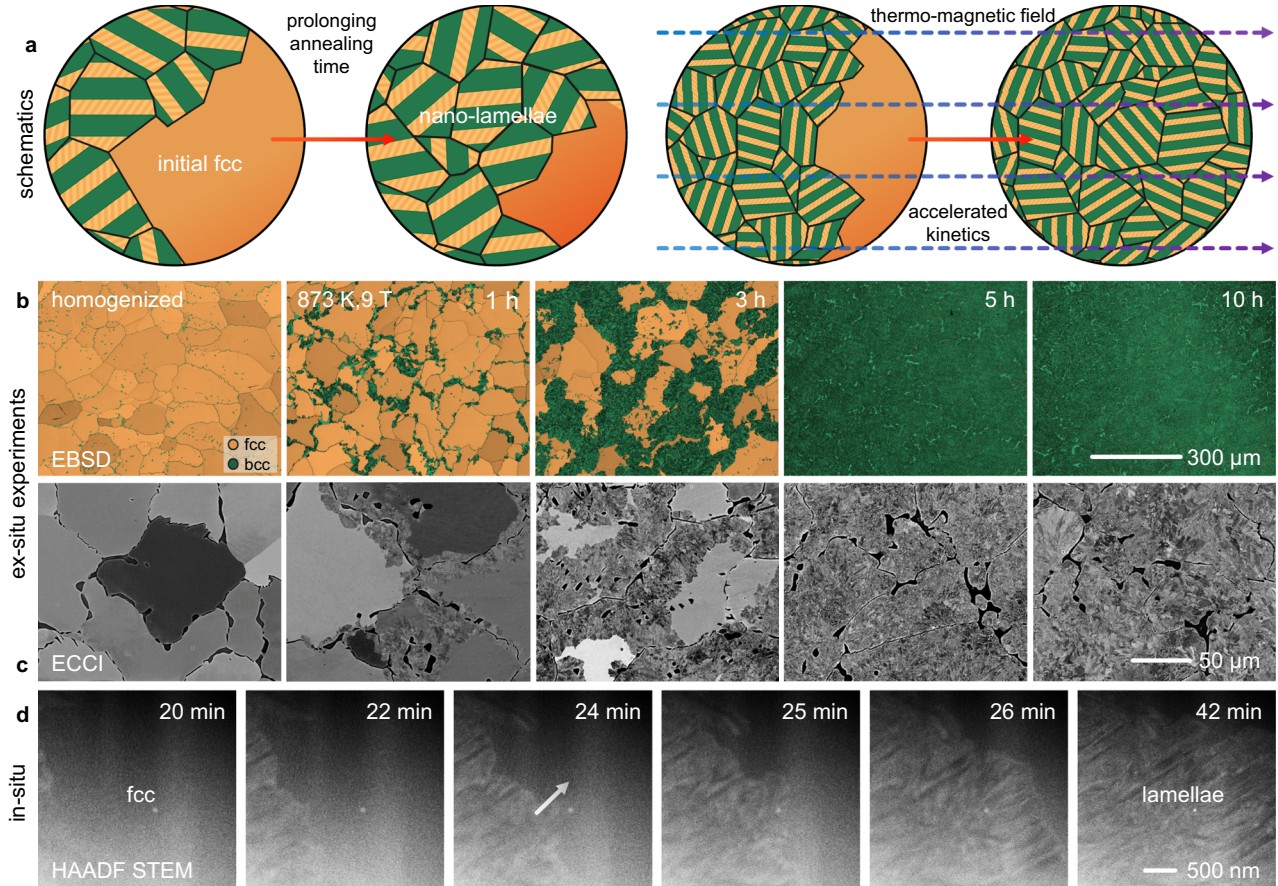

**Fig. 1 | Evolution of eutectoid decomposition in the Co–Fe–Ni–Al MCA.**
**a** Schematic illustration showing the design strategy characterized by harnessing a nano-lamellae decomposition step under simultaneous exposure to an external thermo-magnetic field. An applied external thermo-magnetic field can potentially accelerate the kinetics of phase decomposition and reduce eutectoid colony size

and lamellae spacing. **b** Ex-situ EBSD phase maps and **c** ECCI micrographs showing the phase formation of the bulk material by TMA processing. **d** Time-resolved HAADF-STEM images extracted from in-situ TA experiments showing the formation of nano-lamellae at 873 K.

increased with prolonged TMA time until the remaining fcc phase was completely consumed after 5 h of exposure. The phase decomposition kinetics obtained by the TA treatment are slower than those obtained by the TMA treatment, as it takes longer (6 h) to entirely consume the original fcc phase in the TA-treated magnets. This was also indicated by the difference in the migration rate of the decomposition interface (Supplementary Fig. 2). Figure 1d shows the eutectoid decomposition at the submicrometer scale by in-situ scanning transmission electron microscopy (STEM) analysis at 873 K (Methods). High-angle annular dark-field (HAADF) images revealed the formation and growth of nano-lamellae during TA.

Figure 2a shows the typical hysteresis loops of the MCAs of different thermo-magnetic treatments. The evolution of averaged intrinsic coercivity ($H_c$) and saturation magnetization ($M_s$) values as a function of annealing time is shown in Fig. 2b and Supplementary Fig. 3a, respectively. The $H_c$ and $M_s$ values are averaged from at least three measurements. After 5 h of TMA processing (TMA5), the $H_c$ of the magnet increased by a factor of nearly 40 (~3930%) compared to that of the homogenized condition (H-MCA, "H" referred to as "homogenized"), i.e., from 0.7 kA m⁻¹ to 27.5 kA m⁻¹. This value is 2 times larger than the highest reported $H_c$ value (12.7 kA m⁻¹) in the Co–Fe–Ni–Al system[40,41]. The $M_s$ value increases from 110.5 Am² kg⁻¹ (H-MCA) to 123.8 Am² kg⁻¹ (TMA5) and then decreases to 118.4 Am² kg⁻¹ (10 h of TMA, TMA10). To reveal the improvement in $H_c$ and $M_s$ by the external thermo-magnetic field, the evolution of the magnetic performance as a function of annealing time for the TA-treated material is also presented. The trends for the TA-processed MCAs are similar to those

observed for the TMA-processed MCAs, except that the peak $H_c$ and $M_s$ values are lower and are reached at a later stage, i.e., 19.9 kA m⁻¹ and 120.2 Am² kg⁻¹ after 6 h of TA processing, respectively. To highlight the tuneable magnetic properties of the MCAs, we compare the $M_s$ vs. $H_c$ values with those of commercial magnets and present the magnetic MCAs in an Ashby-type overview plot (Fig. 2c). This comparison showed that the strategy of introducing and tailoring nano-lamellae by TMA treatment allows tuning of the Co–Fe–Ni–Al MCAs from a soft (H-MCA) to a semihard magnetic state (TMA5) and simultaneously increases the $M_s$. The sustainable considerations including alloy cost, carbon and energy footprint of the current alloy are also compared with the commercial semihard magnets (Supplementary Fig. 4). More specifically, the TMA5 magnet has a high $H_c$ value, thus outperforming all the Fe–Co–Ni alloys studied thus far, comparable to those of the Fe–Co–V–Cr alloys with a lower fraction of critical elements Co and Ni and thus lower cost. For instance, the Fe–Co–V–Cr alloys generally contain a high Co content of 49~54 wt.% with $H_c$ of ~28 kA/m. In contrast, the TMA5 magnet contains less amount of Co of ~32 wt.% with comparable $H_c$. In addition, the $M_s$ of the TMA5 magnet (123.8 Am² kg⁻¹) is much higher than that of the Fe–Co–V–Cr alloys (<100 Am² kg⁻¹).

To understand the mechanisms behind the good combination of high coercivity and saturation magnetization of the TMA5 magnet, we studied its microstructure and chemistry from the microscale down to the near-atomic scale (Fig. 3). High-resolution ECC imaging and transmission Kikuchi diffraction analysis (Supplementary Fig. 5) show that the decomposition region consisted of fine eutectoid colonies

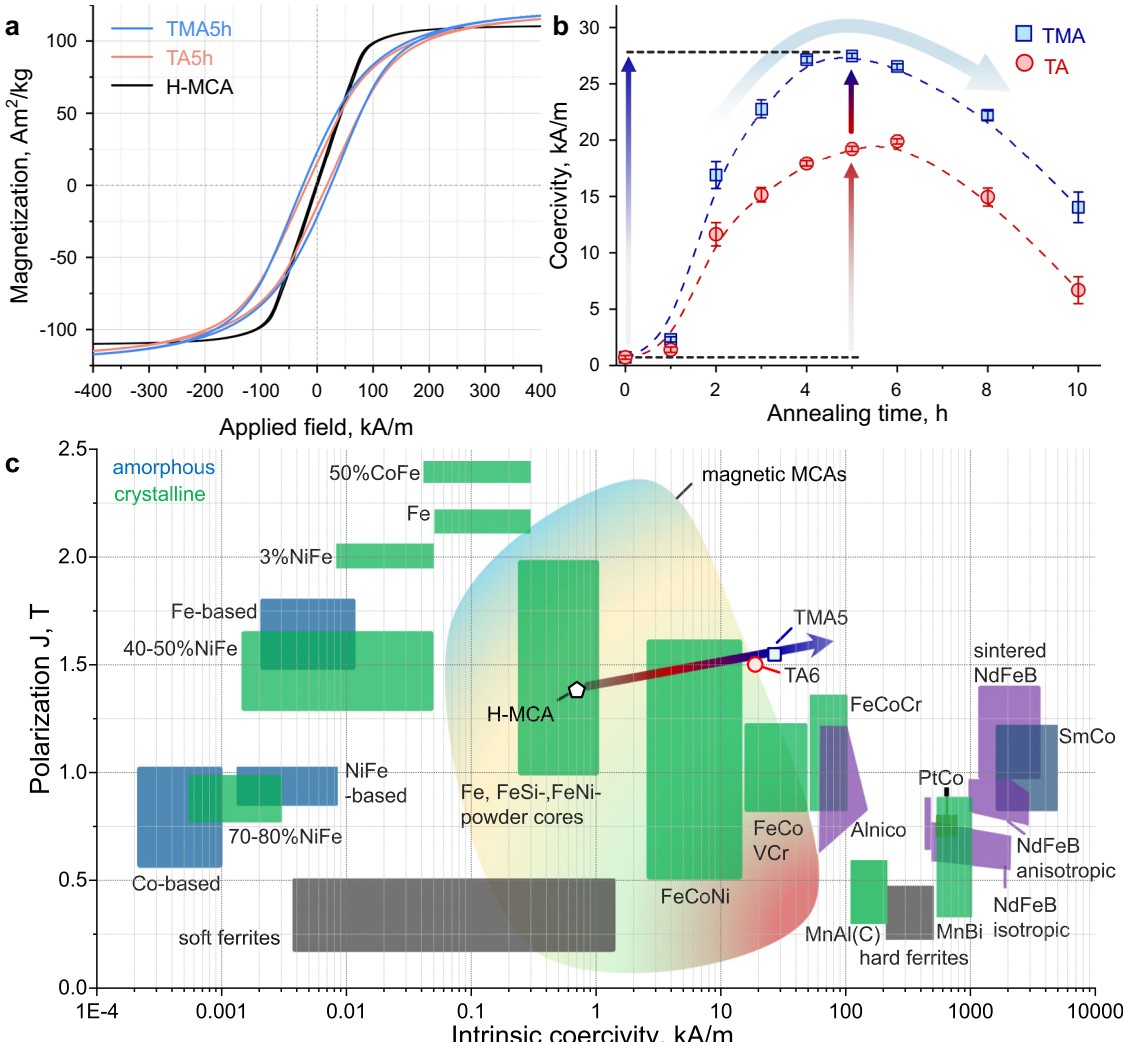

**Fig. 2 | Magnetic performance of the Co–Fe–Ni–Al MCA. a** Room-temperature hysteresis loops of the current MCAs, including H-MCA, TMA5h, and TA5h. **b** Average coercivity ($H_c$) values versus annealing time during the TMA and TA treatments. The error bars are standard deviations obtained from at least three measurements. **c** $M_s$ vs. $H_c$ of the current Co–Fe–Ni–Al magnets, conventional soft magnetic materials, rare earth free and rare earth hard magnetic materials, as well as established magnetic MCAs. TA6, MCA after 6 h of TA processing. Adapted with permission from ref. 1, Wiley.

with an average size of 4.5 ± 3.4 μm. The HAADF-STEM analyses (Fig. 3a) show that the average lamella spacings of the alternating dark and bright layers are 64.7 ± 22.9 nm and 182.2 ± 14.6 nm, respectively. These layers are hereafter referred to as "fine" and "coarse" layers based on their lamellar width as indicated by the arrows. In addition, network structures with much finer widths (9.1 ± 2.3 nm) are distributed in the coarse layer. Through atomically resolved HAADF-STEM micrographs and corresponding fast Fourier transform patterns, both the network structure and the surrounding coarse layer were identified as ordered B2 structures (Fig. 3b). The fine layer is a disordered fcc structure with a uniform distribution of coherently ordered L1$_2$ structures (Fig. 3c). The formation of the hierarchical structure through solid-state phase decomposition results from the competition between severe lattice distortion and chemical ordering tendencies. The chemistry of the nano-lamellae at the near-atomic scale was investigated by atom probe tomography (APT, Fig. 3d–g). Figure 3d shows a three-dimensional reconstruction containing the fine and coarse layers highlighted by one set of interfaces containing 23.0 at.% Co. Figure 3e provides a 10 nm-thick projection obtained from the APT tip, indicated by the black dashed circle. We selected two representative cylinders across the interface between the (a) network

structure and its surrounding coarse layer (blue cylinder) and (b) the L1$_2$/fcc interface (red cylinder) in the fine layer. The corresponding one-dimensional compositional profiles computed along the arrows in Fig. 3e are shown in Fig. 3f, g, respectively. The network structure holds a Ni- and Al-enriched composition of Ni$_{53.9}$Al$_{30.7}$Fe$_{8.3}$Co$_{7.1}$ (at.%), whereas Fe and Co are partitioned into the coarse layer with a stoichiometry given by Co$_{47.2}$Fe$_{43.6}$Ni$_{4.6}$Al$_{4.5}$ (at.%). The compositions of the fcc and L1$_2$ phases were determined to be Ni$_{33.0}$Co$_{31.9}$Fe$_{28.8}$Al$_{6.3}$ and Ni$_{65.7}$Al$_{20.0}$Co$_{7.3}$Fe$_{7.0}$ (at.%), respectively.

**Magnetic structure characterization**

Next, we investigated the magnetic features of the TMA5 magnet to gain insight into the underlying magnetic mechanisms (Fig. 4). We conducted magneto-optical Kerr effect (MOKE) microscopy to characterize the interaction between the magnetic domains and nanolamellae during magnetization and demagnetization (Fig. 4a). Two magnetic domains of different sizes were observed by magnetizing the material from a demagnetized state to an applied magnetic field of 300 mT and then demagnetizing it to −200 mT. First, we observed the nucleation and growth of coarse domains with an average magnetic domain width of 3.6 ± 1.5 μm inside the micron-sized bcc phase

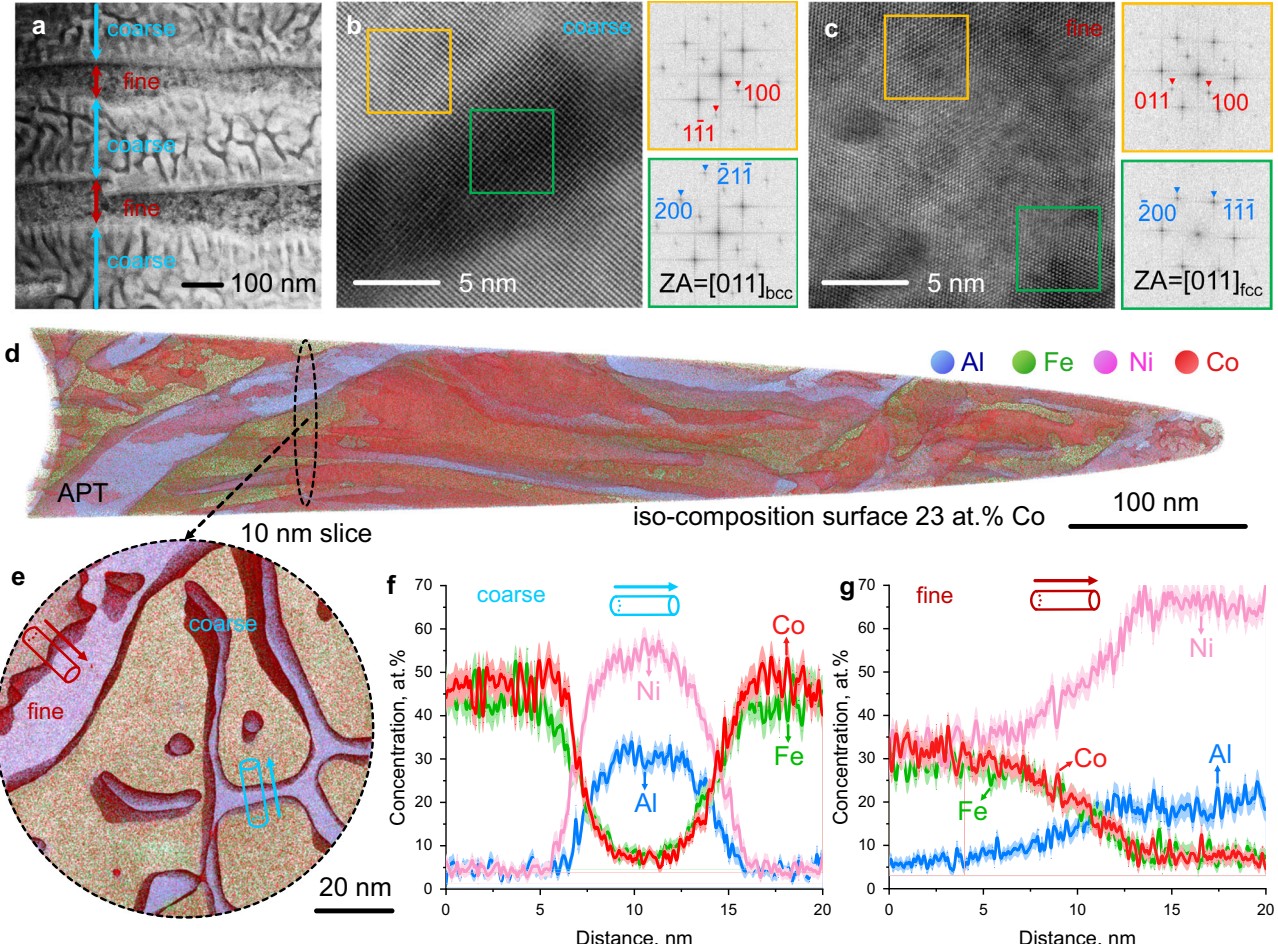

**Fig. 3 | Microstructure of the TMA5 magnet from the micro- to atomic scale.**
**a** HAADF-STEM micrograph showing the coarse and fine layers of eutectoid nano-lamellae. Atomic resolution HAADF-STEM images and corresponding FFT patterns of the (**b**) coarse layer showing that both the low (dark) and high Z contrast (bright) regions are ordered B2 structures and that the **c** fine layer shows an ordered L1$_2$ and fcc structure sharing a coherent interface. ZA, zone axis. **d** APT reconstruction showing the distinct elemental distribution of the nano-lamellae. **e** 10 nm-thick projection highlighted by the interface containing 23.0 at.% Co. The corresponding chemical profiles of the blue and red cylinders with dimensions of $\varphi 5$ nm × 20 nm along the arrows are shown in (**f** and **g**), respectively. The methodology for estimating the error bars is described in the Methods.

fraction. This phase formed during homogenization with a small volume fraction of about 15% (Supplementary Figs. 6 and 7). Second, we observed fine domains uniformly distributed in the eutectoid decomposition region with an average domain width beyond the resolution of optical microscopy, as indicated by the difference in contrast due to the Kerr effect. The domains grow unaffected inside the micron-sized bcc phase but become impeded at the phase boundaries. The correlative magnetic force microscopy (MFM)–atomic force microscopy (AFM) maps (Fig. 4b, c) also allowed us to characterize the domain patterns at higher magnification. Magnetic domains in a maze pattern are observed in the micron-sized bcc phase (Fig. 4b), indicating that magnetic anisotropy extends perpendicular to the sample surface. An enlarged view of the magnetic domains (white frames in Fig. 4b) shows strong magnetic contrast between the fine (bright) and coarse layers (dark) in the demagnetized state when imaged under MFM (Fig. 4c). The corresponding AFM topography shows that the eutectoid region is smooth with a maximum height difference of 9 nm. This finding indicates that the distinctive domain patterns observed in the nano-lamellae are mainly attributed to the difference in magnetization between the alternating layers. Figure 4d, e show the magnetic domain structures under magnetic remanence conditions (0 mT) obtained using Lorentz TEM (LTEM). The alternating bright and dark contrast lines in the defocused

Fresnel images outline the domain walls between the adjacent magnetic domains[51]. Using a series of defocus LTEM images and extrapolating the linear fit to zero for the divergent domain walls (Supplementary Fig. 8) allowed us to estimate the average domain wall width in the micron-sized bcc phase to a range of $43 \pm 6$ nm. This value is typical for soft magnetic materials. In contrast, the enlarged MFM image (Fig. 4e) taken from the nano-lamellae shows that the alternating eutectoid layers act as pinning sites against the movement of the magnetic domain walls.

Since coercivity is an extrinsic property, i.e., a feature largely determined by the interaction of magnetic domain walls with the microstructure, coercivity strongly depends on the distribution of the individual lattice defects and the pinning they exert on the magnetic domain walls. The analysis conducted above indeed showed that the significant increase in the coercivity of the TMA5 magnet during the TMA process was mainly attributed to the strong pinning effect caused by the introduced microstructural defects. More precisely, the microstructure features that contributed to the main pinning effect in the TMA5 material included eutectoid colonies with a small average size of $4.5 \pm 3.4$ μm and nano-lamellae with a fine lamella spacing of $123.5 \pm 18.7$ nm. More specifically, the increase in coercivity until the completion of eutectoid decomposition (5 h for TMA) is correlated with the gradually increasing volume fraction of the nano-lamellae and

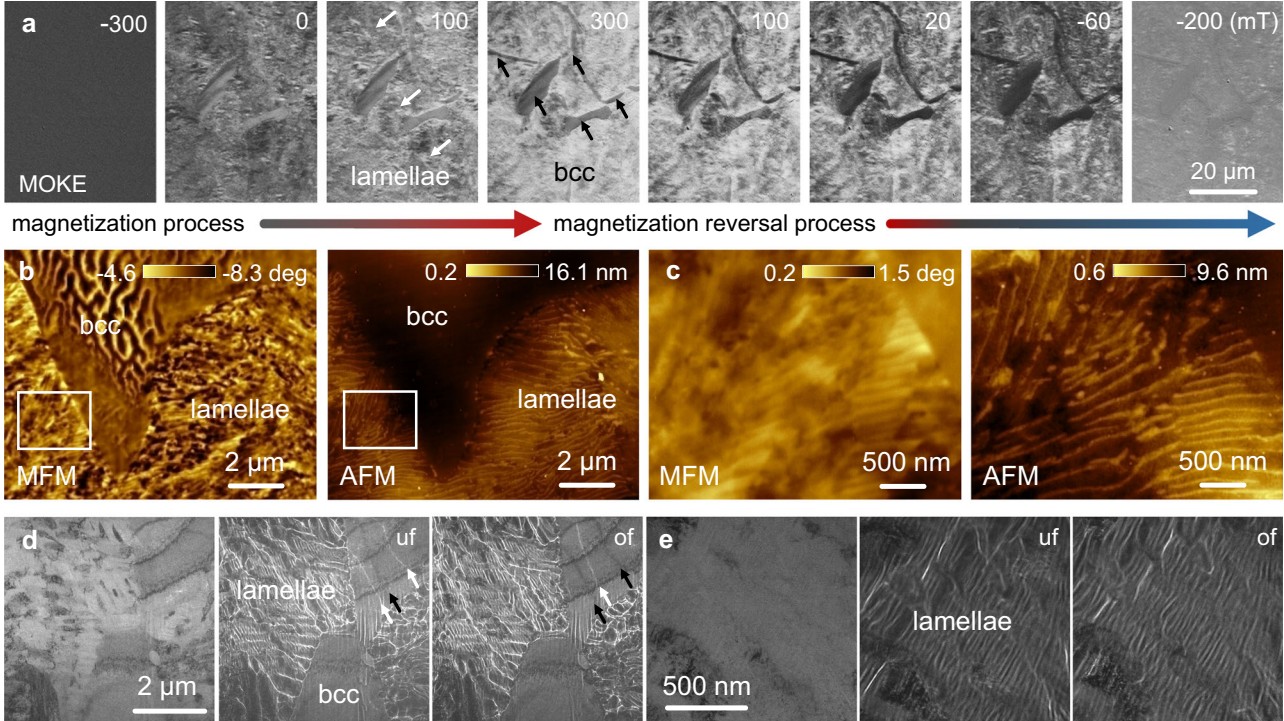

**Fig. 4 | Magnetic feature analysis of the TMA5 magnet. a** Growth of the magnetic domain structure by Kerr-microscopy under different external magnetic fields. **b** Correlative MFM-AFM micrographs showing the magnetic domains. **c** Enlarged views showing the topology and magnetic contrast of the nano-lamellae taken from the white frames in (**b**). Lorentz TEM images showing the (**d**) magnetic domain walls in the Fresnel defocused condition, as indicated by the alternating white and black arrows, and **e**, magnetic microstructure of the nano-lamellae recorded under magnetic remanence conditions at higher magnification. uf, underfocus; of, overfocus.

the corresponding interfacial density (see the statistical analysis in Supplementary Fig. 9). Further annealing leads to decreased coercivity due to the coarsening of colony size and lamellae spacing, e.g., the values are $12.9 \pm 10.1$ μm and $192.6 \pm 36.6$ nm for the TMA10 magnet, respectively. This is driven by a decrease in the total interface energy.

Opportunities for future research lie in targeting material variants with enhanced magnetic performance while preserving their mechanical performance with lower material costs for enhanced multifunctional performance. This can be achieved by tailoring the chemistry, dimension, interlamellar spacing, and number density of the nano-lamellar phases based on the fact that they are the dominant microstructural features contributing to high coercivity and mechanical strength. For instance, increasing the density of the pre-existing heterogeneous nucleus sites (e.g., grain boundary, dislocations) can alter the phase decomposition kinetics and lead to a smaller eutectoid cellular size and inter-lamellar spacing. This can be achieved by increasing grain boundary density via different manufacturing techniques such as additive manufacturing[52]. In addition, the current multi-grained cast condition can be modified towards a stronger grain texture preference (Supplementary Fig. 10) with well-aligned nano-lamellae features for enhanced remanence via additional thermo-mechanical treatment (e.g., plastic forming, cold drawing, directional solidification).

We next studied the intrinsic magnetic properties of the current MCA magnets by in-situ magnetization-temperature investigations (Supplementary Fig. 11). Upon heating from 300 K to 1000 K at a heating rate of 20 K/min, the magnetization of H-MCA drops sharply to a value of only 3.5% of its room-temperature value at 750 K. In contrast, TMA5 retains 37.5% of the room-temperature magnetization at 1000 K. Further analysis of the change in the slope of the curves indicates that the H-MCA magnet contains a ferromagnetic phase with a Curie temperature ($T_c$) of 688 K, while the TMA5 magnet contains two

ferromagnetic phases with higher Curie temperatures, i.e., $Ni_{33.0}Co_{31.9}Fe_{28.8}Al_{6.3}$, $T_c \sim 740$ K; $Co_{47.2}Fe_{43.6}Ni_{4.6}Al_{4.5}$, $T_c > 1000$ K, respectively. We studied the bulk magnetic behaviour of the phases contained in the nano-lamellae (TMA5) by casting these phases into individual ingots using the corresponding nominal chemical compositions derived from APT analysis (Fig. 3). The results showed that the enhanced $M_s$ of TMA5 was mainly due to the formation of a ferromagnetic Co-Fe-enriched B2 phase with high magnetization, i.e., a total magnetic moment of $4.06\mu_B$/f.u., as calculated by density functional theory (Methods), compared with that of the fcc phase in H-MCA before thermo-magnetic annealing, i.e., $1.40\ \mu_B$/f.u.

## Mechanical performance

We applied high-throughput nanoindentation to statistically analyze the hardness of the current MCA material after 3 h of TMA treatment (TMA3). The measured microstructures consisted of micron-sized bcc and nondecomposed coarse-grained fcc, as well as eutectoid nano-lamellae structure, which are typical features observed in the material before and after TMA treatment. To capture the mechanical properties of the different microstructures, an indentation array of $150 \times 150$ indents over a region of 120 μm × 120 μm with an indentation depth of 100 nm and a spacing of 800 nm between indents was constructed at room temperature. Figure 5a shows an EBSD phase map of the TMA3 alloy taken before nanoindentation mapping. Two crystallographic structures were identified, i.e., fcc and bcc. Figure 5b shows an overview of the ECCI micrograph containing 22,500 indents after high-throughput nanoindentation mapping. We assigned each indent to its corresponding microstructure based on the high-resolution ECC image, as shown for a representative region in Fig. 5c. The nano-lamellae are considered here as one phase because the mechanical properties of the finest substructure and phase mixture states, i.e., the eutectoid nano-lamellae contains nanosized fcc and bcc phases (see

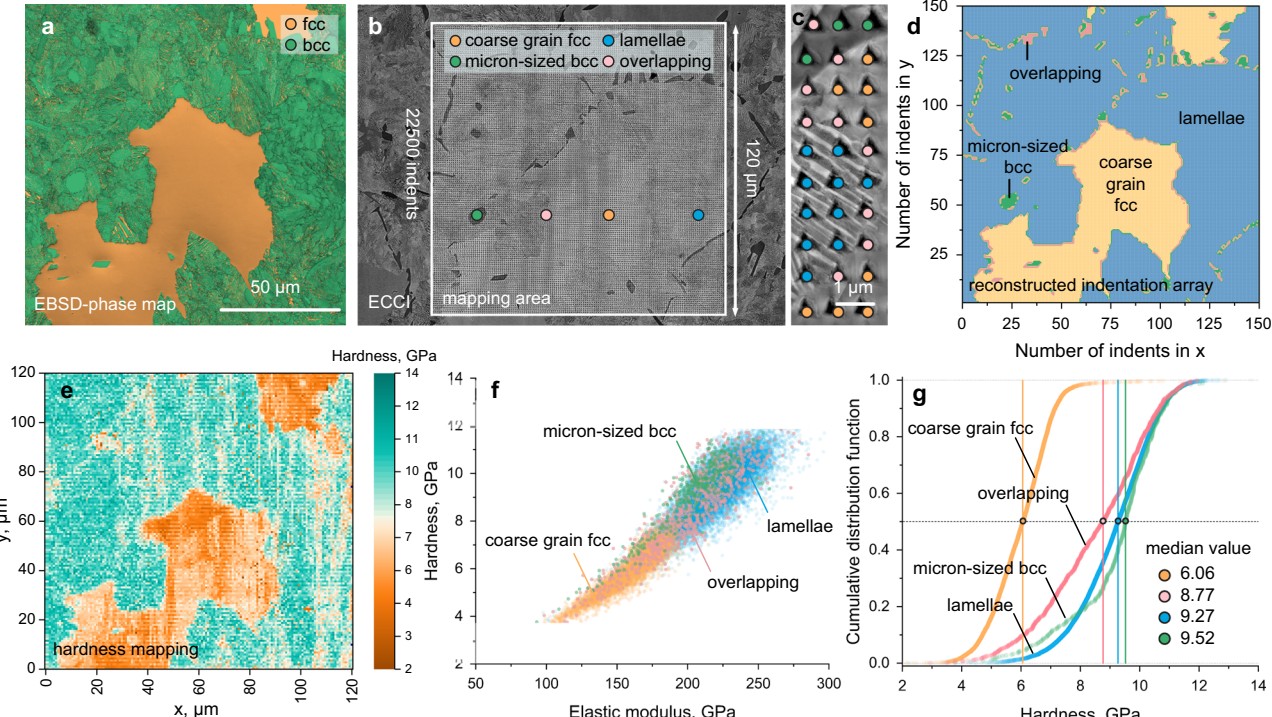

**Fig. 5 | Mechanical performance of the MCA by high-throughput nanoindentation mapping. a** EBSD phase map before indentation. **b** ECCI showing the area of high-throughput nanoindentation containing 22,500 indents. **c** Representative enlarged view showing the method of assigning each indent (indent spacing ~800 nm) to its corresponding microstructure. **d** Reconstructed indentation array showing that the microstructure distribution is identical to that in the white frame in (**b**). **e** Mapping of hardness values obtained from nanoindentation experiments. **f** Cluster distribution of the hardness concerning the elastic modulus values of the undecomposed coarse grain fcc, micron-sized bcc, nano-lamellae, and overlapping region. **g** Corresponding cumulative distribution function of the hardness. The median hardness value of each microstructure was inserted.

Fig. 3a) could not be resolved based on the nanoindentation experiments. The indents located across the different microstructures are referred to as "overlapping" in the following text. The reconstructed indentation array shows the microstructure distribution, as indicated by the different colors (Fig. 5d). The experimentally determined hardness was plotted over the relative coordinate of each indent, resulting in the hardness map shown in Fig. 5e. This distribution matches well with the EBSD and ECCI results. Further hardness/modulus analysis was performed on all the nanoindentation data points to determine the hardness vs. elastic modulus (Fig. 5f). Four hardness/modulus cluster distributions were identified. The hardness and elastic modulus of the nano-lamellae and micron-sized bcc phases are much stronger than those of the undecomposed coarse-grain fcc microstructure. More specifically, the analysis of the cumulative distribution function of the hardness (Fig. 5g) shows that the median hardness values for the nano-lamellae, micron-sized bcc, and undecomposed coarse-grain fcc are 9.27, 9.52, and 6.06 GPa, respectively. The hardness distribution of the overlapping region shows a scatter from ~3 to 12 GPa, covering the scatter range of all the microstructures. The high hardness measured in our experiments is attributed to the indentation size effect[53], which is not unexpected considering the small indentation depth of <100 nm. The accommodation of different microstructures to overall plastic deformation was investigated by nanoindentation experiments to a higher depth (Supplementary Fig. 12). The average hardness values of the fcc and eutectoid lamellae phases are ~3.65 GPa and ~6.81 GPa, respectively. The results show that the hardness values of the major phases are still high, regardless of the indentation size effect. Microstructural analysis of the deeper indents, conducted at a load of 200 mN, revealed the formation of dense slip lines corresponding to the $<110>_{fcc}$ slip systems in the undecomposed

coarse-grained fcc microstructure around the indents. The slip lines stopped at the fcc/micron-sized bcc boundary or at the fcc/nano-lamellae microstructure. In addition, the bulk mechanical performance of the H-MCA and TA-5h alloys were measured by compressive experiments (Supplementary Fig. 13). Both alloys can be plastic compressed up to 50%, indicating their good plasticity. The TA-5h alloy containing eutectoid lamellae shows enhanced yield strength and strain hardening performance as compared to the H-MCA. These results indicate that the current design strategy of introducing nano-lamellae via eutectoid decomposition can significantly harden the magnetic materials.

These considerations show that introducing nano-lamellae with high shape anisotropy through a eutectoid decomposition reaction also enables mechanical hardening of the material. Nano-lamellae are unusual features of advanced magnets but provide an important additional benefit for applications exposed to mechanical loads. The size, density, interface, and chemistry of nano-lamellae tailored by an external thermo-magnetic field must be carefully controlled to increase the number of structural defects to a level that enhances their pinning effect against the motion of both magnetic domain walls and dislocations while avoiding causing mechanical weakness and strain localization that would render them incapable of bearing plastic deformation during processing and service.

In summary, we developed a multicomponent magnet with magnetic properties (coercivity ~27.5 kA m⁻¹, saturation magnetization ~123.8 Am² kg⁻¹) matching those of commercial rare-earth free hard magnets, e.g., Fe–Co–V–Cr alloys, and good mechanical performance in terms of high hardness. We realized this in a Co–Fe–Ni–Al material system by introducing ferromagnetic nano-lamellae with high shape anisotropy by leveraging eutectoid decomposition under an external

thermo-magnetic field. An additional magnetic field significantly accelerates the kinetics of phase decomposition and increases the effective pinning strength by reducing the colony size and lamellae spacing. The nano-lamellae with high average total magnetic moments enhance the saturation magnetization and act as barriers against the motion of both magnetic domain walls and dislocations, thus enhancing the coercivity and mechanical hardness simultaneously. The current design strategy lays the groundwork for developing mechanically strong and magnetically hard multicomponent magnets to operate under harsh service conditions. This is also applicable to other magnetic families where external energy fields or pre-existing defects can modify the microstructural features towards better multi-functionality. Future efforts could target developing variants with enhanced magnetic properties such as higher coercivity and remanence while preserving their good mechanical properties and reducing the dependence on critical elements and thus with lower alloy cost for sustainable electrification. Secondary synthesis using scraps or waste materials with higher impurity contents but lower cost can also be used and considered as raw materials[54], while the potential effect of impurity elements on the microstructure and properties is worth further investigation.

## Methods

### Materials
An alloy ingot with a predesigned nominal composition of $Co_{29.4}Ni_{29.4}Fe_{29.4}Al_{11.7}$ (at.%) was synthesized by vacuum induction melting using pure metallic bulks under a high-purity argon atmosphere. The chemical composition of the cast ingot was determined by wet-chemical analysis as $Co_{28.6}Ni_{28.8}Fe_{30.6}Al_{12.0}$ (at.%). The as-cast ingot with dimensions of 40 mm × 20 mm × 20 mm (length × width × thickness) was then homogenized at 1473 K for 2 h under an argon atmosphere, followed by water quenching. The homogenized material denoted H-MCA ("H", referred to as "homogenized") exhibited a dual-phase structure (fcc+bcc) according to X-ray diffraction analysis (Supplementary Fig. 6). The chemical compositions of the fcc and bcc phases obtained by electron probe microanalysis (Supplementary Fig. 7) were $Co_{29.2}Fe_{31.5}Ni_{28.1}Al_{11.2}$ and $Co_{22.3}Fe_{19.8}Ni_{31.4}Al_{26.5}$ (at.%), respectively.

Further thermo-magnetic annealing was conducted in a physical property measurement system (PPMS, Quantum) at 873 K under a 9 T magnetic field with dwelling times ranging from 1 h to 10 h under a high vacuum atmosphere ( < 0.005 mTor). The heating and cooling rates were 50 K/min, and the magnetic field-varying rate was 1.2 T/min. For comparison, thermomagnetic analysis was performed without the additional magnetic field in the PPMS equipment or the conventional annealing furnace. In addition, bulk ingots of 50 g with identical compositions to those of the disordered fcc phase ($Ni_{33.0}Co_{31.9}Fe_{28.8}Al_{6.3}$, at.%), ordered $L1_2$ phase ($Ni_{65.7}Al_{20.0}Co_{7.3}Fe_{7.0}$, at.%), Ni-Al-enriched bcc phase ($Ni_{53.9}Al_{30.7}Fe_{8.3}Co_{7.1}$, at.%) and Co-Fe-enriched bcc phase ($Co_{47.2}Fe_{43.6}Ni_{4.6}Al_{4.5}$, at.%) in the TMA5 MCA were obtained by arc melting. The compositions of the phases in TMA5 were determined by APT analysis. The ingots were remelted five times under an argon atmosphere to ensure chemical homogeneity.

### Microstructure analytical methods
Multiple techniques were used to characterize the microstructure of the current MCAs. Commercial software (Thermo-Calc) equipped with the high-entropy alloys database (v.4.2) was used for the thermodynamic calculations. X-ray diffraction (XRD) was performed with Co Kα radiation (λ = 1.78897 Å) in an X-ray diffractometer system (D8 Advance A25-X1) at 35 kV and 40 mA. Electron backscatter diffraction (EBSD) was carried out with a scanning electron microscope (Zeiss-Crossbeam) at 15 kV. Electron channeling contrast imaging (ECCI) was performed using a field emission electron microscope (Zeiss-Merlin) at 30 kV. Scanning transmission electron microscopy (STEM)

characterization was performed using a probe-corrected STEM microscope (Titan Themis 60-300) at an acceleration voltage of 300 kV. High-angle annular dark-field (HAADF) micrographs were acquired with a convergence angle of 23.8 mrad. The TEM specimen for in-situ STEM imaging was lifted from an fcc grain close to the <001> zone axis in H-MCA by a focused ion beam (FIB) using a site-specific method[55,56] based on EBSD analysis. The specimen was transferred to a lighting heating/biasing holder (DENS Solutions, Netherlands) and thinned to a final thickness of ~100 nm. The in-situ heating experiment was conducted in the same STEM (Titan Themis 60-300) at 873 K with a heating rate of ~100 K/s. Atom probe tomography (APT) was performed with a local electrode atom probe (LEAP 5000 XR, Cameca) with a pulse frequency of 200 kHz, a pulse energy of 40 pJ, a temperature of 60 K, and a detection rate of 1%. The acquired data were reconstructed and analyzed using commercial software (AP Suite, Cameca). The APT datasets were reconstructed and analyzed using Cameca software (AP suit 6.0). The elemental distribution of the nano-lamellae in the MCA after 10 h of TMA processing (TMA10) determined by APT analysis is shown in Supplementary Fig. 14. The error bars of the one-dimensional compositional profile in the APT analysis were calculated by $2\sigma = \sqrt{\frac{C_i(1-C_i)}{N}}$, where $C_i$ is the composition of each solute $i$ and $N$ is the atom number of the analysis volume.

The magneto-optical Kerr effect (MOKE) for magnetic domain pattern characterization was determined by a Kerr microscope (Axio Imager. D2m, Zeiss). The domain wall motion was observed by varying the magnetic field to within ±0.32 T. Before the measurement, a non-magnetic background was collected as a reference image. Enhanced image contrast was achieved by subtracting the background from the images acquired from different fields using commercial software (KerrLab, evico magnetics GmbH). Correlative magnetic force microscopy (MFM) and atomic force microscopy (AFM) were performed with Bruker Dimension Icon equipment. A silicon cantilever equipped with a pointed tip coated with a Co-Cr ferromagnetic layer (PPP-MFMR, Nanosensors) was used for imaging. The topography was obtained by "tapping mode" in the first scan, while the magnetic structure at the same position was subsequently obtained in a second scan at a constant distance from the surface by adding a specific "lift height" of approximately 50 nm. Lorentz TEM measurements under magnetic field-free conditions (objective lens off, Lorentz mode) were conducted at 300 kV using an image aberration-corrected TEM (FEI, Titan 80-300). Fresnel images were recorded using a direct electron counting detector (Gatan K2 IS) with 4k × 4k resolution using defocus values up to 2 mm.

### Density functional theory (DFT) calculations
Ab initio calculations for achieving the total magnetization of the Co-Fe-enriched B2 phase in the TMA5 material and the fcc phase in the H-MCA materials at 0 K were performed using the exact muffin-tin orbital method[57]. The Perdew–Burke–Ernzerhof exchange–correlation functional[58] was applied for self-consistent calculations. Chemical disorders were simulated using the coherent potential approximation[59]. The s, p, and d orbitals were included in the basis set for solving the one-electron Kohn–Sham equations.

### Magnetic and mechanical response
The magnetic performance was measured using the PPMS system with the vibrating sample magnetometry option. Cuboid specimens 3 mm × 3 mm × 1 mm (length × width × thickness) in length were prepared for the TMA process and measurement. The hysteresis loops were carried out in a magnetic field range of ±1 T with a sweep rate of 50 mT at 300 K. All the specimens were ground to 2500# before measuring the magnetic performance, and at least three specimens were tested for each condition. Ab initio density functional theory calculations were performed using the exact muffin-tin orbital

method[57]. All related calculations were performed in the ferromagnetic state using the Vienna ab initio Simulation Package with projector-augmented wave potentials[60,61], the total magnetic moment of the Fe-Co-enriched B2 phase in the TMA5 magnet and the fcc phase in the H-MCA before thermo-magnetic annealing were calculated to be 1.50 $\mu_B$/f.u., 4.06$\mu_B$/f.u., and 1.40 $\mu_B$/f.u., respectively.

The nanoindentation experiments were conducted at room temperature using an in-situ nanoindenter (FT-NMT04, Femtotools, Buchs, Switzerland) with a standard diamond Berkovich tip on the surface of the specimen. The surface was ground with SiC paper up to 4000 grit and polished with a 30 nm oxide polishing suspension. Before the nanoindentation tests, the frame stiffness and tip area function were calibrated on a reference material (fused silica)[62]. The strain rate was kept constant at 0.025 s$^{-1}$ (equivalent to an indentation strain rate of 0.05 s$^{-1}$) during the test. The elastic modulus and hardness were measured using the continuous stiffness measurement (CSM) method[63] to a maximum depth of 100 nm (nanoindentation mapping) and a maximum load of 200 mN (deep indents to achieve slip lines). To reduce the potential influence of the surface quality caused by the roughness of the sample surface due to mechanical polishing, hardness values > depth of 50 nm were averaged for every indent.

## Data availability

The data generated in this study are provided in the Supplementary Information/Source Data file. Additional data that support the findings of this study are available on request from the corresponding author or the specify the conditions. Source data are provided with this paper.

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

## Acknowledgements

L.H. acknowledges the China Scholarship Council (201906370028). A.K. acknowledges the European Research Council under the European Union's Horizon 2020 Research and Innovation Programme (Grant No. 856538, project "3D MAGiC"). The authors are grateful for the support of Z.Li., W.Lu., R.Xie., funding from the Deutsche Forschungsgemeinschaft (Project-ID 405553726, TRR270) and the European Innovation Council and SMEs Executive Agency (EISMEA) under grant agreement number 101099736 (Pathfinder Open CoCoMag project).

## Author contributions

L.H.: Conceptualization, Methodology, Investigation, Data Curation, Writing—Original Draft, Writing—Review & Editing, Project administration. J.W.: Formal analysis, Data Curation, Investigation, Writing—Review & Editing. N.J.P.: Investigation, Writing—Review & Editing. F.M.: Methodology, Investigation, Data Curation, Writing—Review & Editing. A.K.: Formal analysis, Investigation, Writing—Review & Editing. R.S.: Resources, Writing—Review & Editing. O.G.: Resources, Writing—Review & Editing. D.R.: Writing—Review & Editing, Resources, Supervision.

## Funding

## Competing interests

The authors declare no competing interests.
