## [Transparent Peer Review file · Nature Communications]

Magnetic and mechanical hardening of nano-lamellar magnets using thermo-magnetic fields

Corresponding Author: Dr Liuliu Han

Version 0:

Reviewer comments:

Reviewer #1

(Remarks to the Author)

- What are the noteworthy results?

This paper reports on the accelerating influence of the application of an external field on the development of a complex eutectoid decomposition pattern, i.e., at both a micron and nano metric scale. The development of these dual scale product phase colonies in the microstructure have an effect on the extrinsic magnetic properties, primarily coercivity. The direct comparison of simple thermal annealing to magnetic annealing was very illustrative. They also did a thorough characterization to illustrate this comparison with several types of microstructural and magnetic measurements including in-situ TEM methods (Lorenz), HAADF along with ATP and MFM-AFM. This provided much detailed information on the relationship between microstructure and magnetic properties for this shape anisotropy mechanism of coercivity that has not been studied much, except in the AlNiCo type of magnets that are well referenced. They also could point to consistent behavior with AlNiCo magnets that the coercivity rises when the spacing between interfaces is decreased.

One of the most useful figures in this paper is the Ashby-type plot of a large number of permanent magnet types. It is clear from this plot that the magnetic alloy studied is quite impressive for Magnetization but is rather unimpressive in coercivity, in spite of the magnetic annealing process that is used to modify the final magnetic microstructure, improving the coercivity. It is also notable that the authors did the extra step of identifying the composition of the constituent phases and produced cast alloys of those alloys to provide further understanding of the measurements of the intrinsic properties, primarily the very high value for magnetization. The mechanical property analysis and its relationship to the microstructural features was investigated by phase sensitive nano-hardness measurements. These results revealed a very high hardness (9 GPa) for the nano-scale eutectoid FCC/BCC structure, the BCC phase (well above Martensite phases) and even the (undecomposed) coarse grain FCC phase was not much softer than Martensite at about 6 GPa. One implication of these ultrahard phases (very strong) is that they can be predicted to be quite brittle unless an ultrafine grain size is generated and retained. Unfortunately, this mechanical property characteristic is counter to one of the stated objectives of the study that the resulting magnets should not be "inherently brittle."

- Will the work be of significance to the field and related fields? How does it compare to the established literature? If the work is not original, please provide relevant references.

While the work is original, the authors only studied isotropic (unaligned) permanent magnets, the results are not as significant as they could be. One thing that might improve the resulting magnetic properties (to make them more significant to the field) could come from using some of the lessons learned from recent work on AlNiCo magnets that is referenced. As a starting condition, to achieve a boost in general magnetic properties, e.g., the energy product, the multi-grained cast condition might be possible to convert to a more grain aligned microstructure by either directional solidification or by mechanical rolling (warm or cold) and annealing/recrystallization in the annealed and quenched condition where the FCC phase is dominant (~85vol.%). If this is successful (without cracking) at producing a strong grain texture preference, the subsequent eutectoid decomposition reaction in an external magnetic field might produce a much better aligned 2 phase microstructure and a much-improved remanence. Of course, it seems like the nano-scale lamellar decomposition product phase distribution may be the best at enhancing coercivity, but there is much more research to do on this. Perhaps the possibilities of modifying the eutectoid decomposition structure (with a "draw" annealing process?) might be more easily controlled than the spinodal decomposition process in AlNiCo and such experimental results that are added to this work would be more significant to the field.

- Does the work support the conclusions and claims, or is additional evidence needed?

The starting composition was described as a "multicomponent material" (MCM) and introduced this as a new type of alloy with "a certain potential for realizing multifunctionality." A brief review of the literature reference list shows that the "MCM" of

this paper is more commonly termed a "high entropy alloy" (HEA) or a "complex concentrated alloy" (CCA). This reviewer would prefer that the authors revert to common usage and change all mentions of MCM back to HEA or CCA in a consistent manner. This change will make readers comfortable with this type of alloy and to expect the actual observed behavior of this system

- Are there any flaws in the data analysis, interpretation and conclusions? Do these prohibit publication or require revision? In the Methods section, Figure S1 is incorrectly stated to be a phase diagram, but it should be described as an illustration of the temperature dependence of the phase evolution of the selected alloy that is simulated by the use of Thermo-calc software with the use of the HEA data base that is useful for selection of the annealing temperatures.

-----This only needs revision.

In the Introduction, the "materials design strategy" for the alloy was stated to be due to its "broad solubility range, pronounced solid solution hardening effect, strong magnetic coupling, and hierarchical precipitation reactions, which are all coupled with tunable magnetic properties." This is a logical strategy and perfectly fine for introducing the alloy in this work, but this strategy would be more accurately described as a selection process to identify the alloy from a set of prior publications.

-----This only needs revision.

- Is the methodology sound? Does the work meet the expected standards in your field?

The methodology is sound from the casting process through all of the characterization work (microstructural and magnetic) that was performed and reported.

- Is there enough detail provided in the methods for the work to be reproduced?

There is sufficient detail in the methods section for all of the experimental work to be reproduced.

The exception to the clear illustration of methods comes in the explanation of the highly enhanced Ms of TMA5 where the authors state that it is due to the formation of a ferromagnetic Co-Fe-enriched B2 phase with high magnetization, where the total magnetic moment was calculated by density functional theory, which should be outlined or referenced in Methods.

Reviewer #2

(Remarks to the Author)

The manuscript is clearly written and, from the technical standpoint, it flows well with results also well explained and covered with not much space for improvement in the proposed scope. For the materials science and engineering community the technical content is interesting considering "what" has been done and "how" it has been done.

Specific comments provided below, which should be taken as constructively as possible, address in fact the first question to be asked: "why" it should be done. The manuscript reports a strategy to develop improved rare-earth free PMs, which is certainly of high interest for different reasons (e.g., high price of RE-based magnets, sustainability, ...), focusing on mechanical and magnetic performances. However, some connections/choices/take aways are unclear.

Firstly the authors mention the mechanical failures of PMs; in industry, they are virtually null (no matter the type of PM: Nd-based, Sm-Co, ferrites) compared to common reasons that downtime motors (authors' example in the Abstract) as indicated below. Is it desirable to have PMs with adequate mechanical performance? Yes, but we already do have it, so this argument is not applicable. Second, on the material choice: the manuscript aims to support sustainable electrification as mentioned in the Conclusion, but the system selected uses non trivial quantities of nickel and cobalt that are not the cleanest elements to be mined and processed. Certainly the work focuses and explains the approach proposed and a penalty will always be paid to obtain PMs in its final form, which leads me to the magnetic performance: best coercivity reported is inferior to 30 kA/m. Despite the high saturation, this Hc is of limited applicability. I fully understand that it is not a manuscript reporting a product of any sort, but high-speed motors and generators as indicated by the authors nowadays can have other options to be developed (e.g., Ce-based (not rare-earth-free, but with current developments), Mn-Al(C), Mn-Bi, and others) with superior Hc and closer to be implemented.

Therefore, the "why" question remains unanswered on the specifics truly linking material selection and performance for mentioned applications.

ABSTRACT

(1) The authors mention that PMs are "... often exposed to high mechanical loading conditions during manufacturing and service. ... and fail under such constraints." This is not a fully accurate statement: on the manufacturing side the authors should provide examples (not necessarily in the abstract, but along the text) of what is meant by such comment because, in case of issues during production, factories have developed strategies to minimize waste (if this is the angle pursued; if not, another reason to clarify it). On the service side this is not the case at all: how often do the authors hear that, for instance, high-speed motors and generators (applications mentioned in the abstract) fail because PMs fractured due to mechanical loading? This is a virtually impossible event since PMs are glued onto rotors with appropriate adhesives and motor lifetimes are typically 10+ years not only for industrial space (e.g., interior permanent motors for EVs, fans, pumps, etc.) as well as residential (e.g., outer rotor topology used in washing machines) applications. Certainly PMs must have minimal mechanical performance, which is already achieved today. Therefore, in case the authors want to explore mechanical development as part of research it is certainly a relevant topic, but not using such argument: PMs are not structural components. For the record: in the industrial motors area, a main reason for downtime with motors are bearings.

INTRODUCTION

(1) Page 2 line 43: PMs are used for electrification in general, whether or not the energy source is sustainable/renewable, aiming higher compactness, power density, and obviously efficiency.

(2) As mentioned in the ABSTRACT session, good mechanical performance is already achieved today and if failures do exist in certain applications (the authors must present evidences of these failures so that reader have clearly explained the impacts of such failure) there are issues with the application design.

(3) Page 3 line 66: although the authors provide a technical explanation of why selecting a Alnico-like composition, it would be relevant to understand why other systems were not favored for better magnetic performance. In other words: no hysteresis curves are presented in the manuscript (the ones in the supplemental material should be moved to the paper since it is a PM subject), and the best coercivity achieved is ~ 27.5 kA/m with no indication of remanence (only saturation) along the text ("mixing", then, intrinsic and extrinsic properties, which makes the understanding incomplete). Such H_c , in the space indicated by the authors in the ABSTRACT (high-speed motors and generators) will be of impractical use. In Figure 2c rare-earth free systems of current research (e.g., Mn-Al(C), Mn-Bi), which can present superior performance (not necessarily using the strategy described in the manuscript, I completely understand that) are not even listed. Please provide a high-level view of why pursuing the topic since other systems can present better magnetic performance to Alnico. A final point: since extrinsic properties are listed (H_c), add remanence comments on the manuscript and images updated to tesla.

(4) Page 3 line 91 "... PMs have a limited lifetime and performance due to brittleness": please see comments above.

MAGNETIC STRUCTURE CHARACTERIZATION

(1) Page 9 line 232-234: terms repeating (value typical for soft magnets). Please review.

CONCLUSIONS

(1) Line 349 - 351: I understand what the authors mean when it is mentioned about the approach development for new PMs, but at the same time the rare-earth free system selected uses a non-trivial quantity of cobalt and nickel which, from the sustainability standpoint, is not negligible (for producing metallic Co some info can be found on <https://www.sciencedirect.com/science/article/pii/S2300396018301836>; for Ni: <https://link.springer.com/article/10.1007/s11367-016-1085-x>).

Version 1:

Reviewer comments:

Reviewer #1

(Remarks to the Author)

I agree that it is appropriate to classify the magnets described as semi-hard.

Fig. R.1 is good support for the increase in strength and retained ductility for the highly refined nano-lamellar microstructure after the eutectoid transformation.

Very impressive cold rolling capability exhibited in Fig. R2.

Fig. R3 shows that anisotropy seems to develop in Fig. R3d even after the lowest degree of CR after TMA, which is nice!

Fig. R5d shows the most prevalent nano-lamellar nucleation/growth but am not sure of the alignment of grains (or preferred nucleation sites for the eutectoid) by cold rolling. More study will show this for a follow-up paper, as indicated by the authors!

I agree a lower TMA temperature should improve refinement (Fig. R5), while LPBF may improve situation, but I am not sure.

I appreciate the methods addition for DFT used (and refs.) and am satisfied.

No other comments are needed since all other manuscript additions are well founded and acceptable.

Response to the Reviewers' Report

Manuscript Title: Magnetic and mechanical hardening of nano-lamellar magnets using a thermo-magnetic field

Manuscript number: NCOMMS-24-41164

We thank the editor and reviewers for the valuable suggestions and comments on our manuscript. Our response is structured as follows: The comments from the reviewers are copied below in black and italic font. For each comment, we present a response item and the corresponding manuscript modifications (**green font**). The changes in the amended manuscript are highlighted in **yellow**.

Reviewer #1 (Remarks to the Author):

- *What are the noteworthy results?*

This paper reports on the accelerating influence of the application of an external field on the development of a complex eutectoid decomposition pattern, i.e., at both a micron and nano metric scale. The development of these dual scale product phase colonies in the microstructure have an effect on the extrinsic magnetic properties, primarily coercivity. The direct comparison of simple thermal annealing to magnetic annealing was very illustrative. They also did a thorough characterization to illustrate this comparison with several types of microstructural and magnetic measurements including in-situ TEM methods (Lorenz), HAADF along with APT and MFM-AFM. This provided much detailed information on the relationship between microstructure and magnetic properties for this shape anisotropy mechanism of coercivity that has not been studied much, except in the AlNiCo type of magnets that are well referenced. They also could point to consistent behavior with AlNiCo magnets that the coercivity rises when the spacing between interfaces is decreased.

One of the most useful figures in this paper is the Ashby-type plot of a large number of permanent magnet types. It is clear from this plot that the magnetic alloy studied is quite impressive for Magnetization but is rather unimpressive in coercivity, in spite of the magnetic annealing process that is used to modify the final magnetic microstructure, improving the coercivity.

Response:

Thank you for your strong support and kind appreciation of our systematic work, especially for pointing out that the Ashby-type plot helps to compare the magnetic performance of the current nano-lamellar magnets with established magnetic materials, including soft, semi-hard, and hard magnetic (also termed permanent magnet, PM) materials. We also agree that the coercivity and energy product values of our nano-lamellar magnets are not competitive with conventional PMs. It is therefore indeed instead more pertinent to classify them as semi-hard magnets. Therefore, complying with the reviewers' concerns we revised our manuscript to focus on our design strategy, that is, triggering and refining nano-lamellae via an additional thermo-magnetic field to enhance both mechanical and magnetic performance.

It is also notable that the authors did the extra step of identifying the composition of the constituent phases and produced cast alloys of those alloys to provide further understanding of the measurements of the intrinsic properties, primarily the very high value for magnetization. The mechanical property analysis and its relationship to the microstructural features was investigated by phase sensitive nano-hardness measurements. These results revealed a very high hardness (9 GPa) for the nano-scale eutectoid FCC/BCC structure, the BCC phase (well above Martensite phases) and even the (undecomposed) coarse grain FCC phase was not much softer than Martensite at about 6 GPa. One implication of these ultrahard phases (very strong) is that they can be predicted to be quite brittle unless an ultrafine grain size is generated and retained. Unfortunately, this mechanical property characteristic is counter to one of the stated objectives of the study that the resulting magnets should not be “inherently brittle.”

Response:

We appreciate your positive view on our characterizations and analyses of the magnetic and mechanical properties of the nano-lamellar magnets. It should be noted that the high hardness of the phases measured from the indentation mapping (6 GPa~9.5 GPa) can be partially attributed to the indentation size effect¹, i.e., higher hardness is expected at a smaller indent depth. This is because we kept the maximum indent depth smaller than 100 nm to achieve a high-resolution mapping with an average indent spacing of 800 nm. The main purpose of nanoindentation mapping is to investigate the mechanical performance of the phases with size down to submicron scale. Therefore, we kept the indentation conditions the same: indent depth and strain rate. The reviewer mentioned that the hardness values of the containing phases in our materials are higher than those of martensite, and therefore, the phases might be possibly brittle. However, we cannot compare these hardness values directly to martensite because our sample does not contain a martensitic phase. To address this concern and comply with the reviewer’s hints, we performed additional indentation experiments at a larger indent depth of >1 μm (Fig. S12) to a maximum load of 200 mN, i.e., within a size regime where the indentation size effect becomes insignificant. The average hardness values of the fcc and eutectoid lamellae phases are ~3.65 GPa and ~6.81 GPa, respectively. The results show that the hardness values of the major phases are still high, regardless of the indentation size effect. We also conducted a bulk mechanical compression test on the as-homogenized and annealed alloys, as shown below in Fig. R1, and in the revised manuscript as Fig. S13. Both as-homogenized (H-MCA) and thermal annealed (TA5h) alloys show good plastic formability, i.e., the alloys can be compressed up to 50% engineering strain. In addition, the TA5h alloy containing eutectoid lamellae shows higher compressive yield strength, i.e., an increase in yield strength of ~260 MPa when compared to the H-MCA, indicating a pronounced strain hardening performance. These results show that introducing nano-lamellae with high-shape anisotropy through eutectoid decomposition enables mechanical hardening of the materials without causing brittleness.

Fig. R1. Bulk mechanical performance of the Co-Fe-Ni-Al alloy with different microstructure features. **a** Compressive engineering stress-strain curve of the as-homogenized (H-MCA) and annealed (TA5h) MCAs containing nano-lamellae feature. Two representative curves for each condition are presented. **b** Enlarged view identical to the red frame in (a) showing the enhanced yield strength of the TA5h MCA compared to the H-MCA.

Modifications:

Please see the revised items in the manuscript on page15:

.....The high hardness measured in our experiments is attributed to the indentation size effect⁵², which is not unexpected considering the small indentation depth of <100 nm.....

.....In addition, the bulk mechanical performance of the H-MCA and TA5h alloys was measured by compressive experiments (Fig. S13). Both alloys can be plastically compressed up to 50% engineering strain, indicating their good plastic formability. The TA5h alloy containing eutectoid nano-lamellae shows enhanced yield strength ($\Delta\sigma_y \sim 260$ MPa) and strain-hardening performance compared to the H-MCA.....

• *Will the work be of significance to the field and related fields? How does it compare to the established literature? If the work is not original, please provide relevant references.*

While the work is original, the authors only studied isotropic (unaligned) permanent magnets, the results are not as significant as they could be. One thing that might improve the resulting magnetic properties (to make them more significant to the field) could come from using some of the lessons learned from recent work on AlNiCo magnets that is referenced. As a starting condition, to achieve a boost in general magnetic properties, e.g., the energy product, the multi-grained cast condition might be possible to convert to a more grain aligned microstructure by either directional solidification or by mechanical rolling (warm or cold) and annealing/recrystallization in the annealed and quenched condition where the FCC phase is dominant (~85vol.%). If this is successful (without cracking) at producing a strong grain texture preference, the subsequent eutectoid decomposition reaction in an external magnetic

field might produce a much better aligned 2 phase microstructure and a much-improved remanence. Of course, it seems like the nano-scale lamellar decomposition product phase distribution may be the best at enhancing coercivity, but there is much more research to do on this. Perhaps the possibilities of modifying the eutectoid decomposition structure (with a “draw” annealing process?) might be more easily controlled than the spinodal decomposition process in AlNiCo, and such experimental results that are added to this work would be more significant to the field.

Response:

Thanks for the great and insightful suggestions. We fully agree with you that the previous version of the manuscript only focuses on isotropic conditions, and there is indeed much space to further improve the performance of the current materials by introducing and exploiting crystallographic and the associated magnetic texture/anisotropy. This is based on the fact that the nucleation of the nano-lamellae phases starts from the grain boundaries, which serve as heterogeneous nucleation sites (Fig. 1c). We indeed agree that it can be inferred that the kinetics of phase decomposition and anisotropy can be further tailored by introducing additional nucleation sites and / or trigger specific texture components to achieve finer lamellar spacing and better magnetic performance.

As a specific example along these lines, as correctly mentioned by the reviewer, the current as-cast condition leads to a polycrystalline material with a relatively large crystal size ($>100\ \mu\text{m}$) and this leaves indeed space for further improvement. Other manufacturing and processing methods, such as directional solidification or thermo-mechanical treatment, can achieve a grain-aligned microstructure with potentially better performance.

We have therefore followed the reviewer’s advice and extended the current research by conducting cold-rolling on the as-homogenized (H-MCA) material condition, followed by thermal annealing (TA) and thermo-magnetic annealing (TMA). We first cold-rolled the H-MCA to 20%, 40%, 60%, and 80% of engineering thickness reduction, as shown in Fig. R2. All alloys can be plastically deformed without failure during cold rolling. This indicates their good forming ability, which can undergo complex thermo-mechanical treatment to convert a more grain-aligned microstructure with a strong texture preference for enhanced magnetic properties.

Fig. R2. Macroscopic image of the as-homogenized alloys after different rolling degrees at room temperature.

To understand the effect of pre-deformation-induced microstructure defects on the kinetics of phase decomposition during further thermal-magnetic annealing, we annealed these cold-rolled alloys with different deformation degrees under the same annealing conditions as the current work. The thermo-mechanical treatment parameters and naming of the alloys are shown in Table R1.

Table R1. Thermo-mechanical treatment conditions of different alloys and their ID.

Rolling degree	Annealing time	Annealing temperature	Applied magnetic field	Alloy ID
20%	/	/	/	CR20
20%	5 h	873 K	0 T	CR20-TA
20%	5 h	873 K	9 T	CR20-TMA
40%	/	/	/	CR40
40%	5 h	873 K	0 T	CR40-TA
40%	5 h	873 K	9 T	CR40-TMA
60%	/	/	/	CR60
60%	5 h	873 K	0 T	CR60-TA
60%	5 h	873 K	9 T	CR60-TMA
80%	/	/	/	CR80
80%	5 h	873 K	0 T	CR80-TA
80%	5 h	873 K	9 T	CR80-TMA

The magnetic performance and microstructure characterization are shown in Fig. R3-5. Cold-rolling-induced microstructural defects and texture were observed in all the cold-rolled alloys (Fig. R4). This indeed leads to an anisotropy in the overall magnetic performance, as shown by measuring the hysteresis loops along the rolling direction (RD) and transverse direction (TD), respectively (Fig. R3a). The effect of thermal and thermo-magnetic fields on the phase decomposition is similar to the one described in the previous version of the manuscript. For instance, the coercivity of the CR20 alloy increases from 1.5 kA/m to 8.2 kA/m for the CR20-TA alloy and further increases to 21.6 kA/m for the CR20-TMA alloy. This indicates that the external energy field can modify the microstructural features towards better performance. The alloys with different pre-deformation degrees show different coercivity values after TMA for 5h, indicating the difference in the kinetics of phase decomposition. However, the coercivity values of all the cold rolled then annealed alloys are smaller than that of the TMA5h, which can be attributed to the difference in the aspect ratio, size, dimension of the nano-lamellae phase, and the volume fraction of the decomposed area. For instance, a large volume fraction of the initial fcc structure have not fully decomposed into the eutectoid lamellae for the TA-processed alloys (Fig. R5a), and the nano-lamellae have coarsened due to the accelerated phase decomposition due to the pre-existing defects (Fig. R5b, c). In addition, it can be seen that the deformation-induced microstructure defects act as nucleation sites for phase decomposition, which accelerates the kinetics (Fig. R5d).

Fig. R3. Room-temperature magnetic hysteresis loops of the MCAs after different degrees of cold rolling and thermo-magnetic annealing.

Room temperature hysteresis loops of **a** The MCAs after 20% thickness reduction by cold rolling along the rolling direction (RD) and transverse direction (TD). **b** The MCAs with different degrees of cold rolling from 20% to 80% of thickness reduction. **c** The CR20 MCA followed by thermal annealing (CR-TA) or thermo-magnetic annealing (CR-TMA) and then measured along the RD. **d** Cold-rolled MCAs followed by TMA treatment measured along the RD. All samples for measurement are in the same geometry.

Therefore, a well-tailored magnetic performance is expected to be achieved by further tuning the dwelling time of the TMA processing. We did not include the results mentioned above in the main manuscript since they are not the main focus of the current work, and we wish to take them as our following work with more systematic investigations and characterizations on the difference in phase decomposition kinetics, evolution of microstructure and different properties. Other processing methods to introduce strong texture preference for high texture/anisotropy or tuning the size of the nano-lamellae can also be applied, e.g., using additive manufacturing during processing can provide higher cooling speed and thus lead to finer grain size and higher density of microstructural defects.

Fig. R4. Typical ECCI images of the H-MCA after different degrees of cold rolling. The zoom-in images (bottom line) show the evolution of dislocations.

Fig. R5. BSE images showing the phase decomposition of the nano-lamellae phases. **a** CR40-TA5h MCA containing a relatively large area without initiating phase decomposition. **b** CR60-TA5h MCA containing deformation-induced microstructure defects (microbands). The microbands act as additional nucleation sites for phase decomposition. The nano-lamellae structure of the **c** TMA5h and **d** CR80-TMA5h showing the difference in interlamellar spacing and eutectoid cellular size.

Modifications:

To increase the significance of our work to the magnetic community and related fields, we have added additional discussion items to the revised version of the manuscript to comply with the reviewer and strengthen the paper's message.

Please see the revised manuscript items on page 13:

...Opportunities for future research lie in targeting material variants with enhanced magnetic performance while preserving their mechanical performance with lower materials costs for enhanced multi-functional performance. This can be achieved by tailoring the chemistry, dimension, interlamellar spacing, and number density of the nano-lamellar phases, as these features are the dominant microstructural parameters contributing – when properly tuned – to high coercivity and also to improved mechanical strength. For instance, increasing the density of the pre-existing heterogeneous nucleus sites (e.g., at grain boundaries, triple points and dislocations) can alter the phase decomposition kinetics and lead to a smaller eutectoid cellular size and inter-lamellar spacing. This can be achieved by increasing grain boundary density via different manufacturing techniques, such as additive manufacturing. In addition, the current polycrystalline as-cast microstructure can be modified towards a more pronounced crystallographic texture (Fig. S10) with well-aligned nano-lamellae features for enhanced remanence via additional thermo-mechanical treatment (e.g., plastic forming, cold drawing, directional solidification)...

• *Does the work support the conclusions and claims, or is additional evidence needed?*

The starting composition was described as a “multicomponent material” (MCM) and introduced this as a new type of alloy with “a certain potential for realizing multifunctionality.” A brief review of the literature reference list shows that the “MCM” of this paper is more commonly termed a “high entropy alloy” (HEA) or a “complex concentrated alloy” (CCA). This reviewer would prefer that the authors revert to common usage and change all mentions of MCM back to HEA or CCA in a consistent manner. This change will make readers comfortable with this type of alloy and to expect the actual observed behavior of this system

Response:

We highly appreciate this relevant comment that will help us improve the manuscript. We have fully complied and clarified the reversion to the common usage of “multicomponent alloys.” The original definition of high entropy alloy refers to a single massive solid solution phase structure. Yet, there are multiple phases in the current system, so we use simplified but well-acknowledged terminology, “multicomponent alloys (MCAs),” in the revised version.

Modifications:

We have thoroughly changed all abbreviation items “MCM” into “MCA” in the revised manuscript.

• *Are there any flaws in the data analysis, interpretation and conclusions? Do these prohibit publication or require revision?*

In the Methods section, Figure S1 is incorrectly stated to be a phase diagram, but it should be described as an illustration of the temperature dependence of the phase evolution of the selected alloy that is simulated by the use of Thermo-calc software with the use of the HEA data base that is useful for selection of the annealing temperatures.

-----This only needs revision.

Response:

Thank you for the carefully reading of our manuscript and the corrections.

Modifications:

We thoroughly check the revised manuscript to avoid further flaws in the data analysis, interpretation, and conclusions. Please see the revised Fig. S1:

Fig. S1. Temperature dependence of the phase fraction evolution of the current Co-Fe-Ni-Al MCA.

For the selection of the isothermal heat treatment temperatures, we used the thermodynamics software Thermo-Calc equipped with the High Entropy Alloy database (version 4.2)....

In the Introduction, the “materials design strategy” for the alloy was stated to be due to its “broad solubility range, pronounced solid solution hardening effect, strong magnetic coupling, and hierarchical precipitation reactions, which are all coupled with tunable magnetic properties.” This is a logical strategy and perfectly fine for introducing the alloy in this work, but this strategy would be more accurately described as a selection process to identify the alloy from a set of prior publications.

-----This only needs revision.

Response:

Thank you for your suggestion. We fully agree with the reviewer that clarifying the selection process is important for readers to understand better why we selected the current alloy system. We followed this advice and modified the revised manuscript.

Modifications:

Please see the revised introduction:

.....We selected a $\text{Co}_{28.6}\text{Ni}_{28.8}\text{Fe}_{30.6}\text{Al}_{12.0}$ (at.%) MCA system as a basis for this material development. The ferromagnetic 3d transition elements, i.e., Co, Ni, and Fe, are blended in a near-equiatomic ratio. They form a solid solution matrix with high crystal symmetry (face-centered cubic: fcc) due to its broad solubility range and thus enable a ductile plastic response and leverage high spontaneous magnetization. Adding Al with a large atomic size and negative enthalpy when mixed with Co, Ni, and Fe triggers the formation of precipitates, thus enhancing coercivity and mechanical strength. This enables the targeted hierarchical precipitation reactions, coupled with tunable magnetic properties³⁸⁻⁴⁰

- *Is the methodology sound? Does the work meet the expected standards in your field?*

The methodology is sound from the casting process through all of the characterization work (microstructural and magnetic) that was performed and reported.

Response:

We are most grateful for your careful analysis, positive comments, and helpful recommendations.

- *Is there enough detail provided in the methods for the work to be reproduced?*

There is sufficient detail in the methods section for all of the experimental work to be reproduced.

The exception to the clear illustration of methods comes in the explanation of the highly enhanced M_s of TMA5 where the authors state that it is due to the formation of a ferromagnetic Co-Fe-enriched B2 phase with high magnetization, where the total magnetic moment was calculated by density functional theory, which should be outlined or referenced in Methods.

Response:

Thank you for careful reading and pointing this out, which helped us improve the manuscript. We have added the density functional theory to the Methods in the revised manuscript.

Modifications:

Please see Methods:

Density functional theory (DFT) calculations

Ab initio calculations for achieving the total magnetization of the Co-Fe-enriched B2 phase in the TMA5 material and the fcc phase in the H-MCM materials at 0 K were performed using the exact muffin-tin orbital method⁵⁶. The Perdew–Burke–Ernzerhof exchange–correlation functional⁵⁷ was applied for self-consistent calculations. Chemical disorders were simulated using the coherent potential approximation⁵⁸. The *s*, *p*, and *d* orbitals were included in the basis set for solving the one-electron Kohn–Sham equations.

[56] Vitos, L. Total-energy method based on the exact muffin-tin orbitals theory. *Phys Rev B* 64, 14107 (2001).

[57] Perdew, J. P., Burke, K. & Ernzerhof, M. Generalized Gradient Approximation Made Simple. *Phys Rev Lett* 77, 3865–3868 (1996).

[58] Soven, P. Coherent-Potential Model of Substitutional Disordered Alloys. *Physical Review* 156, 809–813 (1967).

Reviewer #2 (Remarks to the Author):

The manuscript is clearly written and, from the technical standpoint, it flows well with results also well explained and covered with not much space for improvement in the proposed scope. For the materials science and engineering community the technical content is interesting considering "what" has been done and "how" it has been done.

Specific comments provided below, which should be taken as constructively as possible, address in fact the first question to be asked: "why" it should be done. The manuscript reports a strategy to develop improved rare-earth free PMs, which is certainly of high interest for different reasons (e.g., high price of RE-based magnets, sustainability, ...), focusing on mechanical and magnetic performances. However, some connections/choices/take always are unclear.

Response:

We are glad about the strong support and the kind appreciation of the scientific quality of our work. We also thank you for the thoughtful and constructive feedback. These points are critical for improving the clarity and relevance of our manuscript. We are delighted to get the opportunity to strengthen our study by revising the manuscript along the lines you suggested.

Firstly the authors mention the mechanical failures of PMs; in industry, they are virtually null (no matter the type of PM: Nd-based, Sm-Co, ferrites) compared to common reasons that downtime motors (authors' example in the Abstract) as indicated below. Is it desirable to have PMs with adequate mechanical performance? Yes, but we already do have it, so this argument is not applicable.

(1) The authors mention that PMs are "... often exposed to high mechanical loading conditions during manufacturing and service. ... and fail under such constraints." This is not a fully accurate statement: on the manufacturing side the authors should provide examples (not necessarily in the abstract, but along the text) of what is meant by such comment because, in case of issues during production, factories have developed strategies to minimize waste (if this is the angle pursued; if not, another reason to clarify it). On the service side this is not the case at all: how often do the authors hear that, for instance, high-speed motors and generators (applications mentioned in the abstract) fail because PMs fractured due to mechanical loading? This is a virtually impossible event since PMs are glued onto rotors with appropriate adhesives and motor lifetimes are typically 10+ years not only for industrial space (e.g., interior permanent motors for EVs, fans, pumps, etc.) as well as residential (e.g., outer rotor topology used in washing machines) applications. Certainly PMs must have minimal mechanical performance, which is already achieved today. Therefore, in case the authors want to explore mechanical development as part of research it is certainly a relevant topic, but not using such argument: PMs are not structural components. For the record: in the industrial motors area, a main reason for downtime with motors are bearings.

Second, on the material choice: the manuscript aims to support sustainable electrification as mentioned in the Conclusion, but the system selected uses non trivial quantities of nickel and cobalt that are not the cleanest elements to be mined and processed. Certainly the work focuses and explains the approach proposed and a penalty will always be paid to obtain PMs in its final form, which leads me to the magnetic performance: best coercivity reported is inferior to 30 kA/m. Despite the high saturation, this Hc is of limited applicability. I fully understand that

it is not a manuscript reporting a product of any sort, but high-speed motors and generators as indicated by the authors nowadays can have other options to be developed (e.g., Ce-based (not rare-earth-free, but with current developments), Mn-Al(C), Mn-Bi, and others) with superior Hc and closer to be implemented.

Therefore, the "why" question remains unanswered on the specifics truly linking material selection and performance for mentioned applications.

Response:

You are correct that the existing permanent magnets (PMs) have already today adequate mechanical properties for many commercial applications. Insofar we entirely agree but we also like to underpin our suggestion that the current work focuses on specific future product directions where secondary performance, that is, mechanical properties, plays an increasingly critical role in designing advanced magnetic materials, including soft, semi-hard, and hard magnets. This is because there is indeed an increasing practical need for providing magnetic materials with additional more multi-faceted property profiles, e.g., magnetism paired with mechanical load-bearing capacity. In addition, the properties of magnets are highly relative to the microstructure, which in turn is dependent on the manufacturing process. It should be noted that the current magnet manufacturing process is far from ideal regarding waste generation and product variability. The brittleness of intermetallic compounds causes the formation of microcracks during rapid cooling and fracture during machining. For instance, Sm-Co-based alloys have a high machining failure of ~20% because of their low fracture toughness ($1.9\sim 2.9 \text{ MPa m}^{1/2}$)^{2,3}.

We are thankful for the agreement that the main goal of this study—to increase magnetic performance (saturation magnetization and coercivity) and mechanical performance (strength and ductility), as also clearly reflected in our title—*magnetic and mechanical hardening of nano-lamellar magnets using a thermo-magnetic field*, has indeed been achieved. Yet, the questions stand: (1) if such mechanical properties are indeed crucial in magnets, and (2) why go for the current material system considering its relatively low coercivity and remanence values and containing a non-neglectable amount of nickel and cobalt?

As outlined below in more detail, we have fully complied and conducted additional experiments and analysis to underpin our claims that (1) there is indeed a practical and increasing need for providing magnetic materials with this property profile (magnetism paired with high mechanical load-bearing capacity), to enable the coming age of electrification for higher efficiency and more complex manufacturing requirement and that (2) we introduce here a general design approach that is suited to reconcile excellent mechanical properties with tunable magnetic features by external magnetic field in the unexplored compositional space. As compared to the conventional semi-hard magnetic materials, i.e., Fe–Co–Cr–V system, the current magnet (TA5h) modified by additional thermomagnetic treatment shows comparable coercivity with higher saturation magnetization and lower cost, which might find potential applications as semi-hard magnetic materials and (3) there is further room for enhancing the magnetic performance of the current alloys and the current design strategy can also be applied to other magnetic materials.

Any electrical equipment, transformer, or motor designed necessarily with magnetic materials is generally subject to a complex mechanical load exposure profile. As you correctly mentioned, many high-speed motors and generators barely fail due to the fracture of PMs under mechanical loading during service. However, the coming age of electrification requires systems with higher

efficiency and materials with better functional and secondary performance, including mechanical, corrosion, and oxidation resistance under harsh environments. For example, the flywheel energy storage systems can be mass-produced at a reasonable cost with an ultra-fast rotating speed (from 20,000 to over 50,000 rpm in a vacuum enclosure). The targeted rotating speed is 30% faster than the current design. This ultra-fast rotating speed generates high mechanical stress on the containing PMs at the rotor cores induced by the centrifugal forces. Almost all conventional PMs would fail at such ultra-high speeds due to their intrinsically brittle nature, thus limiting the efficiency of the whole system. Therefore, besides the excellent magnetic performance, there is an increasing interest in developing advanced PMs with higher mechanical strength and lower density for the next generation of high-efficiency and sustainable electrification systems.

Another example are advanced electromagnetic brakes (EMBs) systems, which are based on Faraday's electromagnetic induction law as the working principle. This brings about the mechanical safety of PMs that are applied in such system, which previous brakes did not have⁴⁻⁷. Additional dynamic mechanical performance, such as impact toughness and compressive strength, are required, and the corresponding research is limited. Other examples are the PMs applied in the system-related application range for supercapacitors and advanced electric propulsion systems, where PMs are exposed to extreme mechanical stresses due to high rotational speeds, vibrations, and dynamic loads.

New emerging magnetic materials also require good mechanical properties already during manufacturing. The challenge of the entire electrification of our industry is that PM components are clearly developing towards a more complex final product shape. This is motivated by higher efficiency, new designs, less material utilization, more sophisticated imprinted magnetization patterns, and different applications. We agree that factories have developed in-house strategies to minimize waste during production. However, going from simple block magnets to ring segments, easily leads to a waste of 30% and more of the material, sometimes recovered in a slurry. The wastage is significantly increased by the chipping of the magnets during cutting, machining, and polishing and is not to be underestimated to actual mounting in the device, especially when magnetized beforehand. Thus, any improvement in mechanical performance will be of great significance. Once the magnet is safely mounted and glued in the pocket of the motor, we agree with you that they are relatively safe. Yet, for dismantling, reuse, recycling, a safe disassemble, and design for recycling, our approach should be highly relevant. Compared to low-cost and sustainable manufacturing methods by casting and forging, many PMs are synthesized by powder metallurgy that involves complex manufacturing steps, including metal powder synthesis, compaction, sintering, and post-treatment. To meet the final product size and dimensional tolerances, magnets need to be subjected to extrinsic mechanical stress on high manufacturing-related, dynamic, and alternating mechanical loads. Just as all other load-bearing and functional components of a machine or an automobile must be designed to withstand the typical operating loads, this also applies to all electrical components. These parts are the mechanically most vulnerable components in electric automobiles nowadays, as they are very brittle. One reason that these PMs need to be produced by the high-cost powder metallurgy methods is that they are intrinsically brittle, thus creating a cosmos of microstructure defects (cracks, delamination, etc.) in the material in the manufacturing process and cannot be fabricated. For instance, a more complex shape of PM components is required to make the air gap flux density sinusoidal to suppress the air gap flux density harmonic of permanent magnet synchronous motor (PMSM)⁸.

We propose some material systems with good mechanical performance that can be plastically deformed during the synthesis to be able to prevail when rolled, bent, punched, stamped, machined, drilled, grind, and cut, which all are associated with very high mechanical loads, either exposed homogeneously across the bulk or very locally. In addition, these additional microstructure tuning methods can be used to tune the microstructure for enhanced anisotropy that enhances the magnetic performance. For instance, as shown in Fig. R4, we can introduce plastic microbands containing high liner dislocation density by mechanically rolling the alloy. These features can act as additional nucleation sites for the subsequent phase decomposition, thus tuning the mechanical and magnetic performance. This strategy can also be applied to other magnets, such as the Mn-Al system⁹.

More importantly, as you correctly mentioned, the current work focuses on proposing novel strategies to tune the performance of the materials. This is also addressed by reviewer 1, who states that by aligning the initial grain to introduce texture by solidification, rolling, or other advanced methods such as additive manufacturing, we can embrace almost unlimited microstructure design freedom and customize the properties.

We next address other valid points raised by the reviewer regarding why we go for the current alloy system which contains nickel and cobalt with additional environmental and supply-chain concerns, while their magnetic properties are not comparable to some new emerging RE-free PMs.

First, the design idea of multicomponent alloys consisting of multiple principal elements opens up relatively unrestricted chemical composition space of conventional materials. This means that we can further reduce the amount and number of critical elements towards compositionally lean magnets without sacrificing the properties to support sustainable electrification. For instance, guided by the temperature dependence of the phase fraction evolution using thermodynamic calculations (Fig. R6), we reduce the amount of Co and Ni by increasing the amount of Fe while remaining a major fcc structure as the major phase. This leads to a compositionally lean $\text{Fe}_{60}\text{Co}_5\text{Ni}_{25}\text{Al}_{10}$ (at.%) counterpart containing only 30 at.% of critical elements (5 at.% Co+25 at.% Ni) as compared to the current $\text{Fe}_{30}\text{Co}_{30}\text{Ni}_{30}\text{Al}_{10}$ (at.%) alloy which contains 60 at.% of critical components (30 at.% Co+30 at.% Ni). The calculated equilibrium volume fraction of the fcc phase in the compositionally lean alloy is higher than the current alloy system. More importantly, it is achieved at a lower homogenization temperature, that is, 88.6% at 1400 K for the $\text{Fe}_{60}\text{Co}_5\text{Ni}_{25}\text{Al}_{10}$ vs. 85.1% at 1473 K for the $\text{Fe}_{30}\text{Co}_{30}\text{Ni}_{30}\text{Al}_{10}$, respectively. In addition, the $\text{Fe}_{60}\text{Co}_5\text{Ni}_{25}\text{Al}_{10}$ alloy is expected to exhibit an enhanced saturation magnetization due to the high amount of Fe with a high magnetic moment. Although it might still be challenging to compete with the magnetic performance of the $\text{Fe}_{60}\text{Co}_5\text{Ni}_{25}\text{Al}_{10}$ alloy with conventional rare-earth PMs, we believe the magnetic performance regarding coercivity and remanence value can be further improved based on the above-mentioned microstructure tuning strategies, including introducing additional induced anisotropy via aligned grain or texture. This is based on the good mechanical performance of the current material system, which exhibits a major fcc phase as the matrix, thus enabling a ductile plastic forming response.

Fig. R6. Temperature dependence of the phase fraction evolution of the modified $\text{Fe}_{60}\text{Co}_5\text{Ni}_{25}\text{Al}_{10}$ (at.%) MCA with lean compositions.

By comparing the magnetic performance of the current $\text{Fe}_{30}\text{Co}_{30}\text{Ni}_{30}\text{Al}_{10}$ alloy in the Ashby plot (Figure 2c) comparing saturation magnetization against coercivity with the established magnetic materials, the TMA5 magnet has a high H_c value, thus outperforming all the Fe–Co–Ni alloys studied thus far, comparable to those of the Fe–Co–V–Cr alloys. In addition, the M_s of the TMA5 magnet ($123.8 \text{ Am}^2 \text{ kg}^{-1}$) is much higher than that of the Fe–Co–V–Cr alloys ($<100 \text{ Am}^2 \text{ kg}^{-1}$).

While we fully acknowledge the challenges associated with mining and processing these elements, we believe the trade-offs are justified in the context of the multifunctional performance we aim to achieve. Furthermore, ongoing developments in recycling and reuse technologies for cobalt and nickel mitigate some of these sustainability issues. As shown below in Fig. R7a-c and the revised manuscript Fig. S10, by comparing the cost of the current alloy and the modified compositionally lean alloy with the commercial Alnico, Cr–Fe–Co, and Fe–Co–V–Cr semi-hard magnetic materials, the current alloy design has significantly decreased the amount of Co with highest price among the major components. Therefore, current alloys' magnetic and mechanical performance overpasses the conventional Fe–Co–V, and Fe–Co–V–Cr semi-hard materials, yet with fewer critical elements at a lower cost.

Fig. R7. Sustainability considerations, including alloy cost, carbon, and energy footprint of the current MCAs compared to the commercial RE-free magnets. Comparison of concentration and price of the critical elements¹⁰ of **a** commercial hard magnetic Alnico alloy, **b** semi-hard FeCoVCr and **c** CrFeCo alloys and the current FeCoNiAl and compositionally lean variant FeCoNiAl. **d** carbon footprint and energy footprint of the critical elements of commercial RE-free semi-hard and hard magnets.

We also agree that other RE-free magnets, such as Mn-Al(C) and Mn-Bi show at the time of writing still superior H_c . Yet, the current work emphasizes the potential of tunable microstructures and focuses on a strategy of mechanical and magnetically hardening the materials; it would be appropriate to take the TMA5h alloy as a semi-hard magnet, as we postulate in the text. This comparison showed that the strategy of introducing and tailoring nano-lamellae by TMA treatment allows tuning of the Co-Fe-Ni-Al MCAs from a soft (H -MCA) to a semi-hard magnetic state (TMA5) and simultaneously increases the M_s . We believe future iterations of this research involved in optimizing the microstructure and composition by using the strategies reported in our work will result in closing the gap in coercivity and magnetic performance as compared with commercial magnetic materials, however with lower cost and extended functionalities thus broader application ranges.

Modifications:

Along the discussion items laid out above we have correspondingly modified the text, focusing on our intention for a more general design strategy, and added additional discussion regarding material choice, sustainability considerations, and future design efforts throughout the revised manuscript.

Please see the abstract:

High-performance magnetic materials based on rare-earth (RE) intermetallic compounds are crucial for the entire field of energy conversion. However, the high price and supply risks of RE elements require developing inexpensive RE-free magnets, especially for applications where medium performance is sufficient. Another challenge for the future design of RE magnets is that they are inherently brittle and face increasing constraints to cope with high dynamic mechanical loading conditions during service and complex shape design during manufacturing towards high efficiency and sustainability.

Please see the introduction on page 2:

Magnetic materials are key components for the green energy transition due to their role in sustainable wind energy conversion, electromobility, automation, and robotics¹ and provide stable high torque in motor applications². They can be classified into soft, semi-hard, and hard magnetic materials based on their coercivity and energy-product values. Critical rare earth (RE) elements³⁻⁶ are generally required in magnetically hard materials to provide strong spin-orbit coupling leading to high magnetocrystalline anisotropy, which is the basis for a large magnetic hysteresis⁷.....

.....In addition, extensive efforts in designing high-performance magnetic materials have been made to meet the ever-increasing requirements for applications under harsh environmental conditions including high temperature, corrosion, hydrogen embrittlement, and mechanical loading conditions for sustainable electrification¹⁴⁻¹⁷. For example, hard magnets with good mechanical properties are required in advanced flywheel energy storage systems that target an ultra-fast rotating speed of over 50,000 revolutions per minute for high-efficiency and sustainable electrification. This is because high dynamic mechanical and cyclic stresses are generated in the rotating parts and almost all conventional hard magnets fail when exposed to such high rotational speeds, thus limiting the efficiency and durability of the whole system.....

.....Therefore, the strongest hard magnets, e.g., Nd-Fe-B²¹⁻²⁴ and Sm-Co²⁵⁻²⁷ alloys, are generally synthesized by powder metallurgy, which includes a series of energy-, time-consuming and health-risky processing steps, i.e., metal powder production, mixing, compacting and sintering, as compared to the conventional casting and forging. Although near-net shape manufacturing can reduce the need for machining and the loss of materials, high-performance magnetic materials manufactured by adjustable processing with the potential to fill the cost and performance gap of the current magnetic systems are required for efficient and sustainable electrification. It should be noted that the current magnet manufacturing process is far from ideal regarding waste generation and product variability. This is because the properties of magnets are highly relative to the microstructure, which in turn is dependent on the manufacturing process. The brittleness of intermetallic compounds causes the formation of microcracks during rapid cooling and fracture during machining. For instance, Sm-Co-based

alloys have a high machining failure of ~20% because of their low fracture toughness (1.9~2.9 MPa m^{1/2}).

Page 3:

.....We selected a Co_{28.6}Ni_{28.8}Fe_{30.6}Al_{12.0} (at.%) MCA system as a basis for this material development. The ferromagnetic 3d transition elements, i.e., Co, Ni, and Fe, are blended in a near-equiatomic ratio. They form a solid solution matrix with high crystal symmetry (face-centered cubic: fcc) due to its broad solubility range, thus enabling a ductile plastic response and leveraging high spontaneous magnetization because of the strong magnetic coupling from the unpaired electrons. Adding Al with a large atomic size and negative enthalpy when mixed with Co, Ni, and Fe triggers the formation of precipitates, thus enhancing coercivity and mechanical strength. This enables the hierarchical precipitation reactions, coupled with tunable magnetic properties³⁸⁻⁴².....

Page 4:

.....In addition, the anisotropic counterparts containing aligned grains and texture induced by mechanical rolling can help to tune the kinetics of the subsequent phase decomposition of the current alloy system. This indicates the possibilities of further tailoring the shape anisotropy and chemistry of the nano-lamellae structure towards an enhanced magnetic and mechanical performance.....

.....This allows such magnets to withstand severe mechanical loading conditions during service as load-bearing components and provides an additional processing window to enhance induced magnetic anisotropy by plastically deforming the material. In contrast, conventional hard magnetic materials face limited application ranges due to their brittleness. This strategy can also be applied to other magnetic materials that can tolerate inelastic loading without catastrophic failure during manufacturing, such as the Mn-Al system⁴⁸.....

Please see the results and discussion on page 7:

.....Sustainability considerations, including alloy cost, carbon footprint, and energy consumption of the current alloy, are also compared with existing commercial semi-hard magnets (Fig. S4). More specifically, the TMA5 magnet has a high H_c value, thus outperforming all the Fe-Co-Ni alloys studied thus far, comparable to those of the Fe-Co-V-Cr alloys with a lower fraction of critical elements Co and Ni and thus lower production costs. For instance, the Fe-Co-V-Cr alloys generally contain a high Co content of 49~54 wt.% with H_c of ~28 kA/m. In contrast, the TMA5 magnet contains a smaller Co fraction of ~32wt.% with comparable H_c . In addition, the M_s of the TMA5 magnet (123.8 Am² kg⁻¹) is much higher than that of the Fe-Co-V-Cr alloys (<100 Am² kg⁻¹).....

Page 13:

...Opportunities for future research lie in targeting material variants with enhanced magnetic performance while preserving their mechanical performance with lower materials costs for enhanced multi-functional performance. This can be achieved by tailoring the chemistry, dimension, interlamellar spacing, and number density of the nano-lamellar phases, as these features are the dominant microstructural parameters contributing – when properly tuned - to high coercivity and also to improved mechanical strength. For instance, increasing the density of the pre-existing heterogeneous nucleus sites (e.g., at grain boundaries, triple points and

dislocations) can alter the phase decomposition kinetics and lead to a smaller eutectoid cellular size and inter-lamellar spacing. This can be achieved by increasing grain boundary density via different manufacturing techniques, such as additive manufacturing. In addition, the current polycrystalline as-cast microstructure can be modified towards a more pronounced crystallographic texture (Fig. S10) with well-aligned nano-lamellae features for enhanced remanence via additional thermo-mechanical treatment (e.g., plastic forming, cold drawing, directional solidification)...

Please see the conclusions on page 16:

.....Future efforts could target developing variants with enhanced magnetic properties such as higher coercivity and remanence while preserving their good mechanical properties and reducing the dependence on critical elements, thus having lower alloy costs for sustainable electrification. Secondary synthesis using scraps or waste materials with higher impurity contents but lower costs can also be used and considered as raw materials⁵³. The potential effect of impurity elements on the microstructure and properties is worth further investigation.

ABSTRACT

(1) The authors mention that PMs are "... often exposed to high mechanical loading conditions during manufacturing and service. ... and fail under such constraints." This is not a fully accurate statement: on the manufacturing side the authors should provide examples (not necessarily in the abstract, but along the text) of what is meant by such comment because, in case of issues during production, factories have developed strategies to minimize waste (if this is the angle pursued; if not, another reason to clarify it). On the service side this is not the case at all: how often do the authors hear that, for instance, high-speed motors and generators (applications mentioned in the abstract) fail because PMs fractured due to mechanical loading? This is a virtually impossible event since PMs are glued onto rotors with appropriate adhesives and motor lifetimes are typically 10+ years not only for industrial space (e.g., interior permanent motors for EVs, fans, pumps, etc.) as well as residential (e.g., outer rotor topology used in washing machines) applications. Certainly PMs must have minimal mechanical performance, which is already achieved today. Therefore, in case the authors want to explore mechanical development as part of research it is certainly a relevant topic, but not using such argument: PMs are not structural components. For the record: in the industrial motors area, a main reason for downtime with motors are bearings.

Response:

Thank you for the insightful suggestions. These link directly to why the current work should be done. We have also replied to this comment and modified our manuscript correspondingly. Please see the above response and modifications.

Modifications:

We modified the text, focusing on a more general design strategy, and added discussion regarding the material choice, sustainable consideration, and future design efforts throughout the revised manuscript.

Please see the abstract:

High-performance magnetic materials based on rare-earth (RE) intermetallic compounds are crucial for the entire field of energy conversion. However, the high price and supply risks of RE elements require developing inexpensive RE-free magnets, especially for applications where medium performance is sufficient. Another challenge for the future design of RE magnets is that they are inherently brittle and face increasing constraints to cope with high dynamic mechanical loading conditions during service and complex shape design during manufacturing towards high efficiency and sustainability.....

INTRODUCTION

(1) Page 2 line 43: *PMs are used for electrification in general, whether or not the energy source is sustainable/renewable, aiming higher compactness, power density, and obviously efficiency.*

(2) *As mentioned in the ABSTRACT session, good mechanical performance is already achieved today and if failures do exist in certain applications (the authors must present evidences of these failures so that reader have clearly explained the impacts of such failure) there are issues with the application design.*

(3) Page 3 line 66: *although the authors provide a technical explanation of why selecting a Alnico-like composition, it would be relevant to understand why other systems were not favored for better magnetic performance. In other words: no hysteresis curves are presented in the manuscript (the ones in the supplemental material should be moved to the paper since it is a PM subject), and the best coercivity achieved is ~ 27.5 kA/m with no indication of remanence (only saturation) along the text ("mixing", then, intrinsic and extrinsic properties, which makes the understanding incomplete). Such H_c , in the space indicated by the authors in the ABSTRACT (high-speed motors and generators) will be of impractical use. In Figure 2c rare-earth free systems of current research (e.g., Mn-Al(C), Mn-Bi), which can present superior performance (not necessarily using the strategy described in the manuscript, I completely understand that) are not even listed. Please provide a high-level view of why pursuing the topic since other systems can present better magnetic performance to Alnico. A final point: since extrinsic properties are listed (H_c), add remanence comments on the manuscript and images updated to tesla.*

Response:

We cordially thank you for the pertinent comments, which helped us improve the manuscript and clarify the novelty of our material design approach. We fully agree with you and have complied.

Modifications:

We have added the hysteresis loops of the H-MCA, which shows the soft magnetic performance for the initial condition, and the TMA5h one with the highest coercivity and saturation magnetization with a semi-hard magnetic performance in the revised main manuscript as Fig. 2a and as below. We also want to mention that we revised the introduction to avoid overstressing when competing with permanent magnetic performance.

Figure 2. Magnetic performance of the Co-Fe-Ni-Al MCA.

a Room-temperature hysteresis loops of the current MCAs, including H-MCA, TMA5h, and TA5h. **b** Average coercivity (H_c) values versus annealing time during the TMA and TA treatments. The error bars are standard deviations obtained from at least three measurements.

Please see the introduction on page 2:

Magnetic materials are key components for the green energy transition due to their role in sustainable wind energy conversion, electromobility, automation, and robotics¹ and provide stable high torque in motor applications². They can be classified into soft, semi-hard, and hard magnetic materials based on their coercivity and energy-product values. Critical rare earth (RE) elements³⁻⁶ are generally required in magnetically hard materials to provide strong spin-orbit coupling leading to high magnetocrystalline anisotropy, which is the basis for a large magnetic hysteresis⁷.....

.....In addition, extensive efforts in designing high-performance magnetic materials have been made to meet the ever-increasing requirements for applications under harsh environmental conditions including high temperature, corrosion, hydrogen embrittlement, and mechanical loading conditions for sustainable electrification¹⁴⁻¹⁷. For example, hard magnets with good mechanical properties are required in advanced flywheel energy storage systems that target an ultra-fast rotating speed of over 50,000 revolutions per minute for high-efficiency and sustainable electrification. This is because high dynamic mechanical and cyclic stresses are generated in the rotating parts and almost all conventional hard magnets fail when exposed to such high rotational speeds, thus limiting the efficiency and durability of the whole system.....

.....Therefore, the strongest hard magnets, e.g., Nd-Fe-B²¹⁻²⁴ and Sm-Co²⁵⁻²⁷ alloys, are generally synthesized by powder metallurgy, which includes a series of energy-, time-consuming and health-risky processing steps, i.e., metal powder production, mixing, compacting and sintering, as compared to the conventional casting and forging. Although near-net shape manufacturing can reduce the need for machining and the loss of materials, high-performance magnetic materials manufactured by adjustable processing with the potential to fill the cost and performance gap of the current magnetic systems are required for efficient and sustainable electrification. It should be noted that the current magnet manufacturing process is far from ideal regarding waste generation and product variability. This is because the properties of magnets are highly relative to the microstructure, which in turn is dependent on the

manufacturing process. The brittleness of intermetallic compounds causes the formation of microcracks during rapid cooling and fracture during machining. For instance, Sm-Co-based alloys have a high machining failure of ~20% because of their low fracture toughness ($1.9\sim 2.9 \text{ MPa m}^{1/2}$).

Page 4:

.....In addition, the anisotropic counterparts containing aligned grains and texture induced by mechanical rolling can help to tune the kinetics of the subsequent phase decomposition of the current alloy system. This indicates the possibilities of further tailoring the shape anisotropy and chemistry of the nano-lamellae structure towards an enhanced magnetic and mechanical performance.....

.....This allows such magnets to withstand severe mechanical loading conditions during service as load-bearing components and provides an additional processing window to enhance induced magnetic anisotropy by plastically deforming the material. In contrast, conventional hard magnetic materials face limited application ranges due to their brittleness. This strategy can also be applied to other magnetic materials that can tolerate inelastic loading without catastrophic failure during manufacturing, such as the Mn-Al system⁴⁸.....

We have modified by updating the units to tesla and adding other RE-free hard magnetic materials, including Mn-Al(C) and Mn-Bi. Please see below and in the revised manuscript as the new Fig. 2c.

Figure 2. Magnetic performance of the Co–Fe–Ni–Al MCA.

c M_s vs. H_c of the current Co–Fe–Ni–Al magnets, conventional soft magnetic materials, rare earth-free and rare-earth containing hard magnetic materials, as well as established magnetic MCAs. TA6, MCA after 6 h of TA processing.

Page 6:

Fig. 2a shows the typical hysteresis loops of the MCAs with different thermo-magnetic treatments. The evolution of averaged intrinsic coercivity (H_c) and saturation magnetization (M_s) values as a function of annealing time is shown in Fig. 2b and Fig. S3, respectively. The H_c and M_s values are averaged from at least three measurements....

(4) Page 3 line 91 "... PMs have a limited lifetime and performance due to brittleness": please see comments above.

Response:

Thanks for helping us improve the manuscript and for mentioning the main reason for downtime with motors in the industrial motors area is bearings. We have removed this sentence and rewrote the introduction to strengthen the manufacturing and service sides and explain why good mechanical performance is important for future magnetic materials design.

Modifications:

Please see the revised sentence in the introduction:

.....In addition, extensive efforts in designing high-performance magnetic materials have been made to meet the ever-increasing requirements for applications under harsh environmental conditions including high temperature, corrosion, hydrogen embrittlement, and mechanical loading conditions for sustainable electrification¹⁴⁻¹⁷. For example, hard magnets with good mechanical properties are required in advanced flywheel energy storage systems that target an ultra-fast rotating speed of over 50,000 revolutions per minute for high-efficiency and sustainable electrification. This is because high dynamic mechanical and cyclic stresses are generated in the rotating parts and almost all conventional hard magnets fail when exposed to such high rotational speeds, thus limiting the efficiency and durability of the whole system.....

MAGNETIC STRUCTURE CHARACTERIZATION

(1) Page 9 line 232-234: terms repeating (value typical for soft magnets). Please review.

Response:

Thank you for the careful reading. We removed the repeating terms.

Modifications:

Please see the revised sentence:

...This value is typical for soft magnetic materials, i.e., 50 nm to a few micrometres....

CONCLUSIONS

Line 349 - 351: I understand what the authors mean when it is mentioned about the approach development for new PMs, but at the same time the rare-earth free system selected uses a non-trivial quantity of cobalt and nickel which, from the sustainability standpoint, is not negligible (for producing metallic Co some info can be found on

<https://www.sciencedirect.com/science/article/pii/S2300396018301836>; for Ni:
<https://link.springer.com/article/10.1007/s11367-016-1085-x>).

Response:

Thanks for your careful reading. We fully agree that the non-trivial quantity of cobalt and nickel should be addressed for better sustainability. We have revised the text accordingly and also provided future directions, including (1) reducing the dependence on critical elements and cost, (2) manufacturing the magnets with high-impurity raw materials (scraps, waste materials) via secondary synthesis. This can reduce the cost-, energy- and CO₂ consumption significantly, while the effect of impurities on the microstructure, as well as magnetic and mechanical performance is worth further investigation.

Modifications:

Please see the revised sentence:

.....The current design strategy lays the groundwork for developing new mechanically strong and magnetically hard multicomponent magnets to operate under harsh service conditions. This also applies to other magnetic materials where external energy fields or pre-existing defects can be leveraged to modify the microstructure towards novel and improved multi-functional feature combinations. Future efforts could target developing variants with enhanced magnetic properties such as higher coercivity and remanence while preserving their good mechanical properties and reducing the dependence on critical elements, thus having lower alloy costs for sustainable electrification. Secondary synthesis using scraps or waste materials with higher impurity contents but lower costs can also be used and considered as raw materials⁵³. The potential effect of impurity elements on the microstructure and properties is worth further investigation.

References

- [1] Nix, W. D., Gao, H. Indentation size effects in crystalline materials: a law for strain gradient plasticity. *Journal of the Mechanics and Physics of Solids* **46**, 411-425 (1998).
- [2] Ren, L., Hadjipanayis, G.C. and Parvizi-Majidi, A. Fracture toughness and flexural strength of Sm (Co, Fe, Cu, Zr) 7–8 magnetic alloys. *Journal of magnetism and magnetic materials*, **257**, 58-68. (2003)
- [3] Cui, J., Ormerod, J., Parker, D. et al. Manufacturing Processes for Permanent Magnets: Part I—Sintering and Casting. *JOM* **74**, 1279-1295 (2022).
- [4] Li, L., Yang, G.L., Wang, L.Q. Dynamic mechanical characteristics of NdFeB in electromagnetic brake. *Defence Technology*, **19**, 111-125 (2023).
- [5] Shen, J., Qin, X. and Wang, Y., 2018. High-speed permanent magnet electrical machines—applications, key issues and challenges. *CES Transactions on Electrical Machines and Systems*, **2**, 23-33 (2018).
- [6] Liang, S., Shao, X., Que, Y., Guo, B., Bao, H., Tang, G., Yan, X., Bao, J., Yang, L., Qin, L., Shu, K. Recent advances in mechanical properties of sintered NdFeB magnets. *Journal of Alloys and Compounds*, **1003**, 175689 (2024).

- [7] He, T., Zhu, Z., Eastham, F., Wang, Y., Bin, H., Wu, D., Chen, J. Permanent magnet machines for high-speed applications. *World Electric Vehicle Journal*, **13**, 18(2022).
- [8] C. Liu, Y. Xu, J. Zou, G. Yu and L. Zhuo, Permanent magnet shape optimization method for PMSM air gap flux density harmonics reduction, *CES Transactions on Electrical Machines and Systems*, **5**, 284-290, (2021).
- [9] Keller, T., Gurau, G., Baker, I., Severe plastic deformation of Mn-Al permanent magnets. *Materialia*, **38**, 102251 (2021).
- [10] Torrubia, J., Valero, A. & Valero, A. Energy and carbon footprint of metals through physical allocation. Implications for energy transition. *Resour. Conserv. Recycl.* **199**, 107281 (2023).